# Importance of potassium ions for ribosome structure and function revealed by long-wavelength X-ray diffraction

Alexey Rozov[1,2,3,4,5,11], Iskander Khusainov [1,2,3,4,6,10,11], Kamel El Omari [7,8], Ramona Duman[7,8], Vitaliy Mykhaylyk [7], Marat Yusupov[1,2,3,4,6], Eric Westhof[9], Armin Wagner [7,8] & Gulnara Yusupova[1,2,3,4]

The ribosome, the largest RNA-containing macromolecular machinery in cells, requires metal ions not only to maintain its three-dimensional fold but also to perform protein synthesis. Despite the vast biochemical data regarding the importance of metal ions for efficient protein synthesis and the increasing number of ribosome structures solved by X-ray crystallography or cryo-electron microscopy, the assignment of metal ions within the ribosome remains elusive due to methodological limitations. Here we present extensive experimental data on the potassium composition and environment in two structures of functional ribosome complexes obtained by measurement of the potassium anomalous signal at the K-edge, derived from long-wavelength X-ray diffraction data. We elucidate the role of potassium ions in protein synthesis at the three-dimensional level, most notably, in the environment of the ribosome functional decoding and peptidyl transferase centers. Our data expand the fundamental knowledge of the mechanism of ribosome function and structural integrity.

[1] Institut de Génétique et de Biologie Moléculaire et Cellulaire, 67404 Illkirch, France. [2] Centre National de la Recherche Scientifique, UMR7104, 67404 Illkirch, France. [3] Institut National de la Santé et de la Recherche Médicale, U1258, 67404 Illkirch, France. [4] Université de Strasbourg, 67404 Illkirch, France. [5] RiboStruct, 15 rue Neuve, 67540 Ostwald, France. [6] Institute of Fundamental Medicine and Biology, Kazan Federal University, Kremlyovskaya street 18, Kazan 420008, Russia. [7] Diamond Light Source, Harwell Science and Innovation Campus, Chilton, Didcot OX11 0DE, UK. [8] Research Complex at Harwell, Rutherford Appleton Laboratory, Harwell Oxford, Didcot OX11 0FA, UK. [9] Université de Strasbourg, CNRS, Architecture et Réactivité de l'ARN, UPR 9002, 15 rue René Descartes, F-67000 Strasbourg, France. [10] Present address: EMBL Heidelberg, Meyerhofstraße 1, 69117 Heidelberg, Germany. [11] These authors contributed equally: Alexey Rozov, Iskander Khusainov. Correspondence and requests for materials should be addressed to A.R. (email: ar@ribostruct.com) or to A.W. (email: armin.wagner@diamond.ac.uk) or to G.Y. (email: gula@igbmc.fr)

Metal ions play key roles in a broad range of cellular processes[1–3]. Typically, alkali and alkaline earth metals, such as $Na^+$, $K^+$, $Mg^{2+}$, $Ca^{2+}$ are critical for the stability, proper folding and functioning of RNA and proteins[4–7], whereas transition metals are also involved in catalysis of redox reactions ($Fe^{2+}$ or $Cu^{2+}$)[8,9] or act as Lewis acids in enzyme active sites ($Zn^{2+}$)[10,11]. In the specific case of RNA, cations facilitate the dense folding arrangements of the negatively charged phosphate backbone of RNA molecules while enabling various cellular functions: gene expression (messenger (m-) RNAs and transfer (t-) RNAs), gene regulation (small nuclear, micro and small interfering RNAs), enzymatic activity (e.g., ribozymes), or resistance to pathogenic and parasitic invaders, observed in Eukarya[12,13].

Functions and structures of biomolecules evolved in intracellular environments with $K^+$ and $Mg^{2+}$ among the predominant cations. Unsurprisingly, while both of these ions contribute to the stability of various RNA structures[14–18], together they demonstrate a more pronounced synergistic effect[19]. These two ions demonstrate significant differences in properties: $Mg^{2+}$ is a small ion (ionic radius 0.72 Å)[20] with high charge density and strong preference of octahedral coordination (coordination number 6), while $K^+$ is larger (ionic radius 1.51 Å)[20], less charged, leaning towards higher coordination numbers (8–12). This precludes their competition and expands the variety of environments and modes of possible interactions of these cations, which is particularly crucial for macromolecular machines.

One such machine is the ribosome, the largest and the most abundant RNA-containing macromolecular complex in cells, ranging in size from 2.5 MDa in bacteria to 4 MDa in higher organisms. Ribosomes are conserved in all kingdoms of life: they are composed of rRNA and proteins unequally distributed among two asymmetric subunits (small and large subunits, 30S and 50S, respectively in bacteria)[21]. Ribosomes perform protein synthesis upon subunit association and interaction with mRNA and tRNA ligands; their structure and function strongly depend on the presence of divalent (mainly $Mg^{2+}$, $Zn^{2+}$) and monovalent (mainly $K^+$, $NH_4^+$) cations[22–25]. Magnesium is the most characterized cation, its importance for ribosome activity was described by pioneers of ribosome research. Lack of $Mg^{2+}$ in growth medium for *E. coli* induces ribosome degradation[26]. In vitro studies demonstrated that $Mg^{2+}$ concentrations below 1 mM cause 70S ribosome subunit dissociation followed by unfolding[27,28]. However, magnesium is not the sole component responsible for proper ribosome activity. Early studies demonstrated that polyamines, particularly spermidine or spermine, can compensate for $Mg^{2+}$ ions for optimum protein synthesis in in vitro translation systems[29–35]. The highest rate of protein synthesis in vitro, however, is achieved in the presence of $Mg^{2+}$, polyamines and monovalent cations ($K^+$/$NH_4^+$) together[36–38]. In addition, magnesium alone is insufficient to recover ribosome sedimentation profiles after treatment with high concentrations of EDTA due to the loss of other required ions[39]. Similarly, the complete substitution of $Mg^{2+}$ by polyamines leads to inactivation and loss of integrity of ribosomal subunits in *E. coli*[40,41]. Polyamines are known to associate stably and abundantly with ribosomes[42] but are very rarely detected in structural studies and even then ambiguously[43], hence it was proposed that most of polyamine binding sites are differentially occupied in a stochastic manner[44]. The other key players of ribosome activity and stability are monovalent ions. In the absence of $K^+$ ions for example, mammalian ribosomes irreversibly lose their poly-Phe polymerizing activity[45], while *E. coli* ribosomes dissociate into subunits upon exposure to very high $K^+$ concentrations[46] or moderate $Na^+$ concentrations[47]. Thus, none of the individual components,

cations or polyamines, can entirely substitute for each other, and efficient translation by the ribosome can only be achieved by correct concentrations and balance between them.

Despite the vast biochemical data regarding the importance of metal ions for effective ribosome performance[25] and the increasing number of ribosome structures solved by X-ray crystallography or cryo-electron microscopy, the identification of metal ions within the ribosome remains elusive due to methodological limitations. Therefore, in the majority of ribosome models derived from conventional data collection used for X-ray structures, metal ions are usually assigned as magnesium—the best-known RNA-stabilizing atom. Consequently, the local chemical environment of the metal ions was interpreted from the point of view of octahedral coordination.

Anomalous X-ray diffraction is a very well established tool to determine and localize ions in three-dimensional structures[48,49]. Every chemical element displays a characteristic set of absorption edges in the X-ray range, corresponding to the binding energies of electrons (K, L, M electron shells corresponding to K-, L-, M-edges). The anomalous signal from atoms of the element under investigation changes drastically across its absorption edge. Hence, peaks in the anomalous difference Fourier map from measurements on the high-energy side, which are not present in data on the low-energy side, reveal the atomic positions of the anomalous scatterers. The majority of synchrotron beamlines for macromolecular crystallography are optimized for the 6–17.5 keV X-ray range[50]. However, to detect and measure the anomalous signal from potassium around its K-edge ($E = 3.608$ keV) access to lower energies is necessary. The long-wavelength beamline I23 at Diamond Light Source is currently the only synchrotron beamline for macromolecular crystallography covering the energy range around the potassium K-edge. Experiments at long wavelengths have a number of obstacles to overcome: mainly large diffraction angles and absorption from air in the beam path, the sample mount, solvent around the crystal and the crystal itself. Beamline I23 has been designed to address these challenges by operating in a vacuum environment with a multi-axis goniometer and a large semi-cylindrical area detector[51].

Here we demonstrate experimental identification and localization of potassium ions within the full 70S ribosome structure using diffraction data collected at long wavelengths, below ($\lambda = 3.542$ Å, $E = 3.5$ keV) and above ($\lambda = 3.351$ Å, $E = 3.7$ keV) the potassium K absorption edge. We present crystal structures of two 70S ribosomal functional complexes with mRNA and tRNAs, representing two distinct stages of translation: initiation and elongation. The 70S ribosome complex in the initiation state contains initiation tRNA$^{fMet}$ paired with the AUG codon in the peptidyl-site (P-site) with the aminoacyl-site (A-site) vacant. In the complex modeling the elongation state, three tRNA$^{Phe}$ are paired with UUU codons in the A-, P- and exit (E-) sites. Our findings provide insights into the role of metal ions in two ribosome active sites, the decoding and peptidyl transferase centers. We demonstrate how $K^+$ (but not $Mg^{2+}$) coordinates mRNA within the decoding center in order to maintain the correct frame position during the elongation state. We also localize potassium ions that are required for subunits association and stabilization of tRNAs, rRNAs, and r-proteins. These results shed light on the role of metal ions for the ribosome architecture and function, thereby expanding our view on fundamental aspects of protein synthesis.

## Results
**Metal ions assignment**. We reinvestigated the structure of the *Thermus thermophilus* 70S ribosome in two different functional states, modeling the initiation stage (further referred to as initiation complex or IC) and elongation stage (further referred to

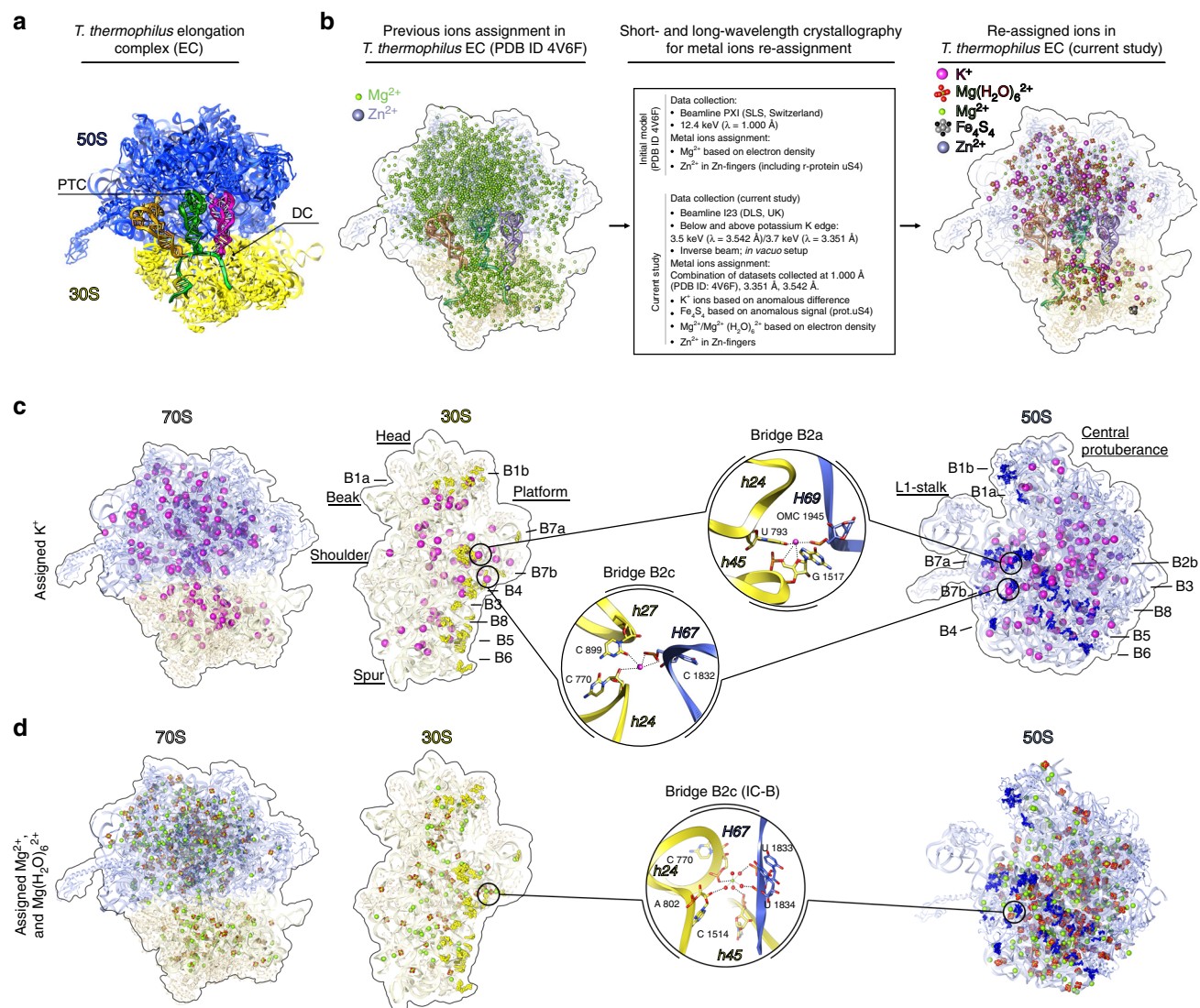

**Fig. 1** Metal ions assignment in *T. thermophilus* 70S ribosome on the example of elongation complex. **a** *Thermus thermophilus* 70S EC represents the elongation state of the ribosome that contains poly-U mRNA with SD sequence and three cognate tRNA$^{Phe}$ in the A-, P- and E-sites. Parts of the central protuberance and the 30S head are omitted for clarity. Abbreviations used: PTC peptidyl transferase center, DC decoding center, SD Shine-Dalgarno. **b** *Thermus thermophilus* 70S elongation complex (PDB ID 4V6F) was used as an initial model for ions re-assignment. The same complex was crystallized and data were collected at beamline I23 (Diamond Light Source, UK) at two different wavelengths (3.542 and 3.351 Å, which correspond to energies below and above the K-absorption edge of potassium). Electron density from three independent datasets collected at different wavelengths (1.000 Å (PDB ID 4V6F), 3.542 and 3.351 Å) were used to build an atomic model, which contained 211 experimentally distinguished K$^+$ ions, 334 Mg$^{2+}$, 251 Mg(H$_2$O)$_6$$^{2+}$, 1 Zn$^{2+}$ and 1 Fe$_4$S$_4$ cluster in place of 3255 Mg$^{2+}$ and 3 Zn$^{2+}$ ions assigned previously. **c**, **d** Distribution of K$^+$ and Mg$^{2+}$ ions within 70S ribosome and each individual subunit. Potassium ions are coordinated in intersubunit bridges B2a and B2c (**c**). One magnesium ion is coordinated in B2c bridge (in the IC magnesium ion has slightly different position and coordinated via water molecules) (**d**). OMC stands for 2'-O-methylcytosine. K$^+$ ions colored in magenta, Mg$^{2+}$ ions in green, solvent shell shown in red, Zn$^{2+}$ in dim gray, in Fe$_4$S$_4$ cluster in iron colored in gray and sulfur in black. Small ribosomal subunit (30S) parts colored in yellow, large subunit (50S) components in blue. Intersubunit bridges are shown in bright yellow and bright blue. For clarity, ions within ribosome are shown in sphere representation with increased van der Waals radius. Additionally, tRNA and mRNA ligands are omitted from the interface of 30S subunit in **c** and **d**

as elongation complex or EC) of translation (Fig. 1a)[52,53]. The initiation complex was reconstituted from empty 70S ribosomes, a 27-nucleotide-long mRNA comprising the Shine-Dalgarno sequence with an AUG codon and a poly(A) extension and tRNA$^{fMet}$ (Supplementary Fig. 1a). In this complex, we found tRNA in the P and E sites; the presence of tRNA in the E site can be explained by the high excess of tRNA used in crystallization. The elongation complex was reconstituted from empty 70S ribosomes, a 60-nucleotide-long mRNA containing the Shine-Dalgarno sequence followed by a poly(U) tail and tRNA$^{Phe}$$_{GAA}$

(Supplementary Fig. 1b). tRNA$^{Phe}$$_{GAA}$ was found in the three sites binding tRNA: the A, P and E sites.

The diffraction data were collected above and below the potassium K-edge, at 3.7 keV (λ = 3.351 Å) and 3.5 keV (λ = 3.542 Å), respectively. K$^+$ ions were assigned in positions showing peaks in the anomalous difference Fourier map above the absorption edge, but not in the corresponding map below the edge, and overlapping with positive peaks in the omit $F_o-F_c$ map, generated from deposited diffraction data (λ = 1 Å) for the published structures[53]. The 70S model of the EC (PDB ID 4V6F)

contained 3255 $Mg^{2+}$ (including the instances of $Mg(H_2O)_6^{2+}$ modeled as $Mg_7$). We have instead identified, 211 $K^+$, based on anomalous data, 334 $Mg^{2+}$ ions and 251 $Mg(H_2O)_6^{2+}$ (Fig. 1b–d). In the 70S model of the IC (PDB ID 4V6G) 1439 $Mg^{2+}$ ions were modeled, which this study re-assigned as 127 $K^+$, 189 $Mg^{2+}$, and 237 $Mg(H_2O)_6^{2+}$ ions. Of course, the ions modeled as $Mg^{2+}$ also should be fully or partially hydrated but relatively high displacement factors or relatively low occupancies prevent modeling of additional water molecules. Moreover, low site occupancy and high atomic displacement factors might have dampened the anomalous signal and prevented us from assigning more $K^+$ ions in place of some of $Mg^{2+}/Mg(H_2O)_6^{2+}$.

Our 70S structures contain 2 ribosomes per asymmetric unit and the exact number of assigned ions varies between the two ribosomes in the asymmetric unit. Below we use for reference one of the ribosomes from the elongation complex, where most of the $K^+$ ions could be assigned (unless stated otherwise). The pairwise comparison of $K^+$ composition in all four ribosomes is shown in Supplementary Table 1. Despite the strong implication of $K^+$ ions in the 70S subunit dissociation, only two $K^+$ ions were found to support intersubunit bridges, namely bridge B2a and bridge B2c, whereas the others were rather uniformly distributed all over the ribosome (Fig. 1c). The majority of $K^+$ ions play a role in the stabilization of RNA through phosphate backbones or via coordination to exocyclic groups on stacked nucleotides. Among others, we have identified several $K^+$ ions that form part of functional sites (the decoding and peptidyl transferase centers), stabilize tRNA ligands, or preserve rRNA–protein interactions. The interacting partners and putative coordination spheres of $K^+$ ions discussed in the text and presented on the figures are summarized in Supplementary Data 1. The preferred coordination of $K^+$ ions in the structures reported in this work were found to be predominantly square anti-prismatic or bi-capped square anti-prismatic (Supplementary Fig. 2) as opposed to octahedral for $Mg^{2+}$ ions. Almost all of the identified $K^+$ ions had been previously assigned as $Mg^{2+}$ or $Mg(H_2O)_6^{2+}$ except 4 in EC and 28 in IC. In addition, our long-wavelength data allowed us to confirm the presence of a $Fe_4S_4$ cluster, which was first assigned by the group of T. Steitz[54], bound to the Zn-finger of the ribosomal protein uS4.

**Potassium ions in the 70S decoding center.** We have closely analyzed the decoding center—the essential ribosomal functional site responsible for correct matching of mRNA codon and aminoacyl-tRNA (aa-tRNA) anticodon during translation. Here, we reassigned two magnesium ions as potassium. One of which, present in both IC and EC (Fig. 2a), stabilizes the ribosomal elements of the decoding center, regardless of the presence of A-tRNA. This $K^+$ ion is coordinated to the conserved nucleotides C518, G529 of the 16S rRNA (we use *E. coli* numbering of rRNA nucleotides throughout the manuscript as well as in the deposited PDB models) and amino acids Pro45, Asn46 of the universal ribosomal protein uS12 (Fig. 2a). The coordination has square antiprismatic geometry with coordination number 8 (Fig. 2b). The second $K^+$ ion interacts with the mRNA in the A-site and helps orienting the third codon base for proper base pairing with anticodon residue 34 (consequently, it was identified only in the EC) (Fig. 2a). This ion is coordinated to the nucleotide (+6) of mRNA (third position of the A-codon, see Supplementary Fig. 1), C518 and G530 in *anti* conformation. The coordination has bi-capped square antiprismatic geometry with coordination number 10 (Fig. 3b). In the absence of A-tRNA, as seen in the IC, the mRNA is displaced farther from helix 18 and nucleotide G530 adopts a *syn* conformation as was shown for other A-tRNA-free ribosomal complexes[53,55] and, thus, no $K^+$ ion is bound in this

region (Fig. 2a, Supplementary Movie 1). Additionally, distance/geometry-based analysis confirms the inability of $Mg^{2+}$ or $Mg(H_2O)_6^{2+}$ to bind within these two pockets (Fig. 2b, c).

**Potassium ions along the messenger RNA path.** As was shown by the previous studies[53,56,57], the mRNA path on the ribosome takes two bends, leading to the formation of a sharp kink between A- and P-codons (A/P kink) and a kink between P- and E-codons (P/E kink) in the mRNA chain (Fig. 3a–c). These kinks of the mRNA together with neighboring 16S rRNA nucleotides form negatively charged pockets that are highly favorable for binding either metal cations or water molecules. In our structure of the EC, we have localized a $K^+$ ion in the P/E kink, which appears to be coordinated through its solvent shell according to the distances in the pocket (Fig. 3b). Previously, this ion, which participates in the network of contacts formed between the $ms^2i^6A37$ modification in the P site $tRNA^{Phe}$ and ribosomal elements surrounding the mRNA P/E kink, resulting in the anchoring of P-site tRNA, was assigned as magnesium[53]. In the A/P kink pocket we did not detect any anomalous signal and the size of the pocket is not favorable for coordination of $K^+$, suggesting that the site is occupied by another ion/molecule interacting only with anionic phosphate oxygens (likely to be $NH_4^+$) (Fig. 3c). We observed electron density in this pocket only in the elongation complex. In fact, summarizing our previous studies[53,58–62], we can point out the inconsistent appearance of the ligand-related density peaks in mRNA kinks. Therefore, we suggest that cations or solvent molecules play a secondary role in the formation of kinks, while the primary is performed by the structure of 16S rRNA and ribosomal proteins[53].

**Metal ions that stabilize other elements of the ribosome.** The step-by-step advance of the ribosome along mRNA, accommodation and release of tRNA during translation require multiple rearrangements of ribosomal subunits relative to each other. A number of regions on the interface of ribosomal subunits, called intersubunit bridges, help to keep the ribosome intact and at the same time ensure its dynamics. Intersubunit bridges were first visualized as immediate contacts in low-resolution cryo-EM studies[63,64]. Then, at least 12 individual intersubunit bridges were identified in the first crystallographic study of functional 70S ribosome complexes[65]. Some of these bridges were suggested to be supported by metal ions. In our structures, we have identified one $K^+$ ion coordinating to elements of the bridge B2a (h24-h45-H69), whereas one $K^+$ and one $Mg^{2+}/Mg(H_2O)_6^{2+}$ ion are coordinated to the constituent elements of the bridge B2c (h24-h27-H67) (Fig. 1c, d, Supplementary Data 1).

$K^+$ ions are also found to stabilize the structure of tRNAs bound to the ribosome (Supplementary Fig. 3a). The A-site tRNA, for instance, contains a $K^+$ ion in the anticodon stem loop (ASL), which presumably supports its accommodated state in the decoding center. The P-tRNA, aside from fixation by ribosomal components, is additionally stabilized by $K^+$ and $Mg^{2+}$ ions. One $K^+$ is coordinating the interaction between its D-stem and H69 of the 23S rRNA (Supplementary Fig. 3A, circle b), while two other maintain the internal structure of P-tRNA in the T-loop and D-stem. As expected, the E-site tRNA is the least stabilized and no ions were found in this region.

**Metal ions in ribosomal proteins and rRNA.** The convoluted three-dimensional folding of the ribosomal RNA implies juxtaposition of nucleotides that are located far apart in the primary structure. Therefore, many metal ions, including potassium, localize in such regions to neutralize charge density and stabilize these nucleotides. The most representative example of such

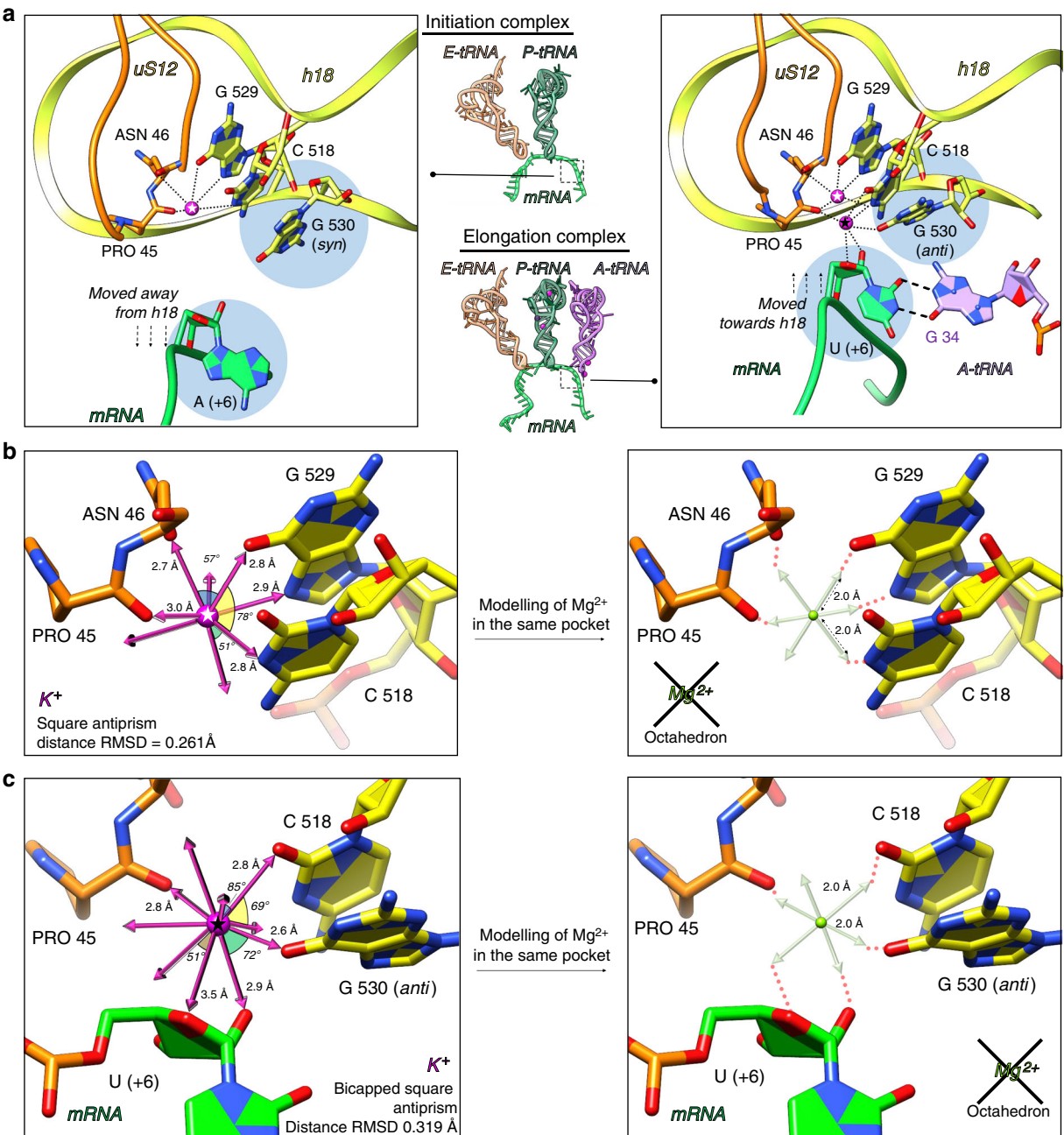

**Fig. 2** Localization of potassium ions in the 70S ribosome decoding center. **a** Structural rearrangements of the decoding center and its stabilization by potassium ions upon binding of A-tRNA. In the initiation complex (left), only one K$^+$ ion conserves the architecture of decoding center through coordination with C518 and G529 of h18 and amino acid residues Pro45 and Asn46 of protein uS12. The mRNA in the initiation complex in the absence of A-tRNA is shifted away from h18, while G530 adopts energetically unfavorable *syn* conformation. In the elongation complex (right), in contrast, mRNA is moved towards h18 in order to form base pairing with the A-tRNA. In this scenario, an additional K$^+$ ion is involved in the stabilization of codon–anticodon interaction via coordination through C518, G530 (in anti-conformation), Pro45 and U(+6) ribose. **b**, **c** The best fitting coordination geometry was estimated to be square antiprism (coordination number 8) with an RMSD of 0.261 Å for the five identified coordinating atoms positions for the "first" K$^+$ ion in the decoding center (**b**, left), and bi-capped square antiprism (coordination number 10) with RMSD of 0.319 Å for the five identified coordinating atoms positions for the "second" K$^+$ ion in the decoding center (**c**, left). In silico modeling shows that Mg$^{2+}$ ion does not fit into these binding pockets due to its distance and geometry constrains (**b**, **c** right). 16S rRNA elements are shown in yellow, nucleotides A/U(+6) and G530 are highlighted by light blue circles. Contacts between K$^+$ and ribosomal components are shown in round dash, U(+6)-G34 base pair is marked by long dash lines, two K$^+$ ions are marked with white star ("first" K$^+$) and black star ("second" K$^+$)

convolution is PTC, where peptide bond formation occurs. We assigned 30 K$^+$ ions around and inside the PTC (Fig. 4a–c). Seven ions are located next to the inner shell of the PTC, and the others coordinate remote A and P loops responsible for orienting the -CCA ends of tRNA substrates. Another composite domain of the

large subunit is the central protuberance—it comprises 5S rRNA wrapped by distant 23S rRNA helices and r-proteins. This region is supported by 12 K$^+$ ions, three of which stabilize the 5S rRNA (Supplementary Fig. 3b) and the others coordinate H38, H83, H84 of 23S rRNA with proteins uL5 and uL16. On the small subunit,

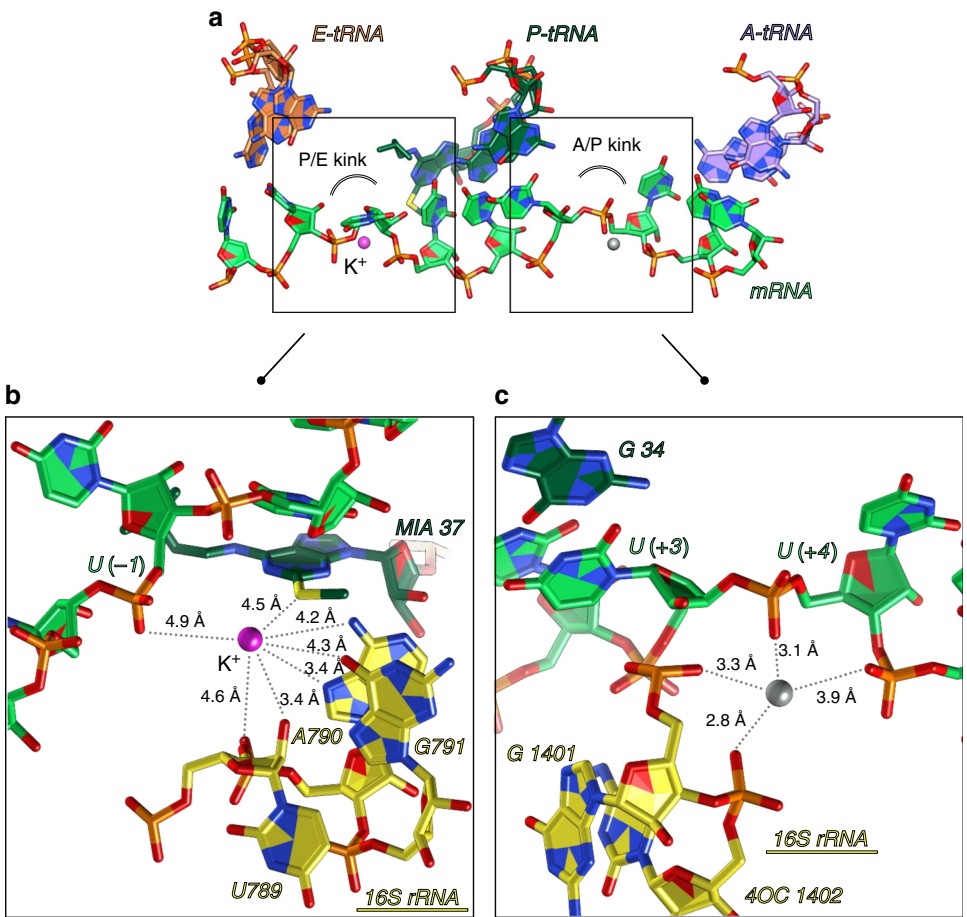

**Fig. 3** Metal ions in the mRNA path in elongation complex. **a** Along its path in the ribosome, mRNA gets distorted between A- and P- sites (A/P kink) and between P- and E-sites (P/E kink). Together with 16S rRNA nucleotides, these distortions represent negatively charged pockets, favorable for occupation by metal ions like $K^+$ or $Mg^{2+}$. **b** A $K^+$ ion was identified in the P/E kink of the elongation complex. According to the distances, this $K^+$ ion is most probably hydrated and may interact with 2-methylthio-N6-isopentenyladenosine (MIA) modification in the P-tRNA$^{Phe}$ and U(-1) in mRNA. **c** The cation identified in A/P kink was assigned as $Mg^{2+}$ in the elongation complex, however, we include the possibility that it can be a different ion (probably an ammonium ion). Color code: colors are as in Fig. 2 with unidentified ion shown in gray

we found $K^+$ ions in the 16S rRNA helix h18 that forms a part of the decoding center (Fig. 5a–c). One of the few and probably the main site, that maintains position of the head on the body of the 30S subunit, helix h28, is strengthened by a $K^+$ ion (Fig. 5d). Helix h44, one of the principal helices of 16S rRNA and a part of the decoding center also contains coordinated $K^+$ ions (Fig. 5e–f).

A number of distant nucleotides that are stabilized by potassium ions are found to be in stacking interaction with G–G as the preferred pattern. Some of those contacts are reminiscent of the interactions between potassium ions and O6 of guanines observed in G-quadruplexes[66,67]. Notably, in the 70S ribosome, we observed 51 $K^+$ ions coordinating G–G stacking through phosphate oxygen backbone atoms (Supplementary Fig. 4a), nucleotide exocyclic groups, like O6 of G (Supplementary Fig. 4b, c) or ribose hydroxyl group and base atom (Supplementary Fig. 4d). We have divided these $K^+$ ions into two types: Type I stabilize two consequent G stacking (Supplementary Fig. 4a, b), Type II are responsible for stabilization of stacking of distant G nucleotides (Supplementary Fig. 4c, d). An extended list of $K^+$ ions that stabilize distant rRNA nucleotides and G-stacking in both subunits can be found in Supplementary Data 1.

Metal ions like $Mg^{2+}$ and $K^+$ are mostly known to interact with RNA components of the ribosome. However, we identified several $K^+$ ions associated with the following ribosomal proteins:

bS6, bS20, uL2, uL3, uL4, uL5 and uL16 (Fig. 6). Most of these ions were found in local turning loops where the carbonyl oxygen atoms of the polypeptide backbone form a negatively charged pocket favorable for $K^+$ ions. Potassium ions that interact with proteins bS20 and uL2 are newly identified ions, and are not replacing previously assigned magnesium.

Additionally, we have localized 12 $K^+$ ions that stabilize the binding of proteins to rRNAs. Of those, 5 $K^+$ atoms coordinate to the 16S rRNA and proteins uS11, uS13, uS14, bS20, and 7 $K^+$ atoms coordinate to the 23S rRNA and proteins uL2, uL4, uL15, bL28 (Supplementary Fig. 5).

## Discussion

The folding of RNA structures requires the presence of counter ions. Large macromolecular complexes which contain nucleic acids necessitate correspondingly large numbers of various metal ions. Improvement of data collection in X-ray crystallography and cryo-EM over the last decades[68,69] has led to more detailed maps of 70S ribosomes, revealing density peaks tentatively attributed to metal ions. A multitude of technical limitations has prevented empirical identification of the nature of these ions and they were generally assigned as magnesium. Magnesium was chosen since it is the best-known RNA-stabilizing counter ion, and ribosomes tolerate only a very narrow concentration range during purification and in vitro experiments. At the same time,

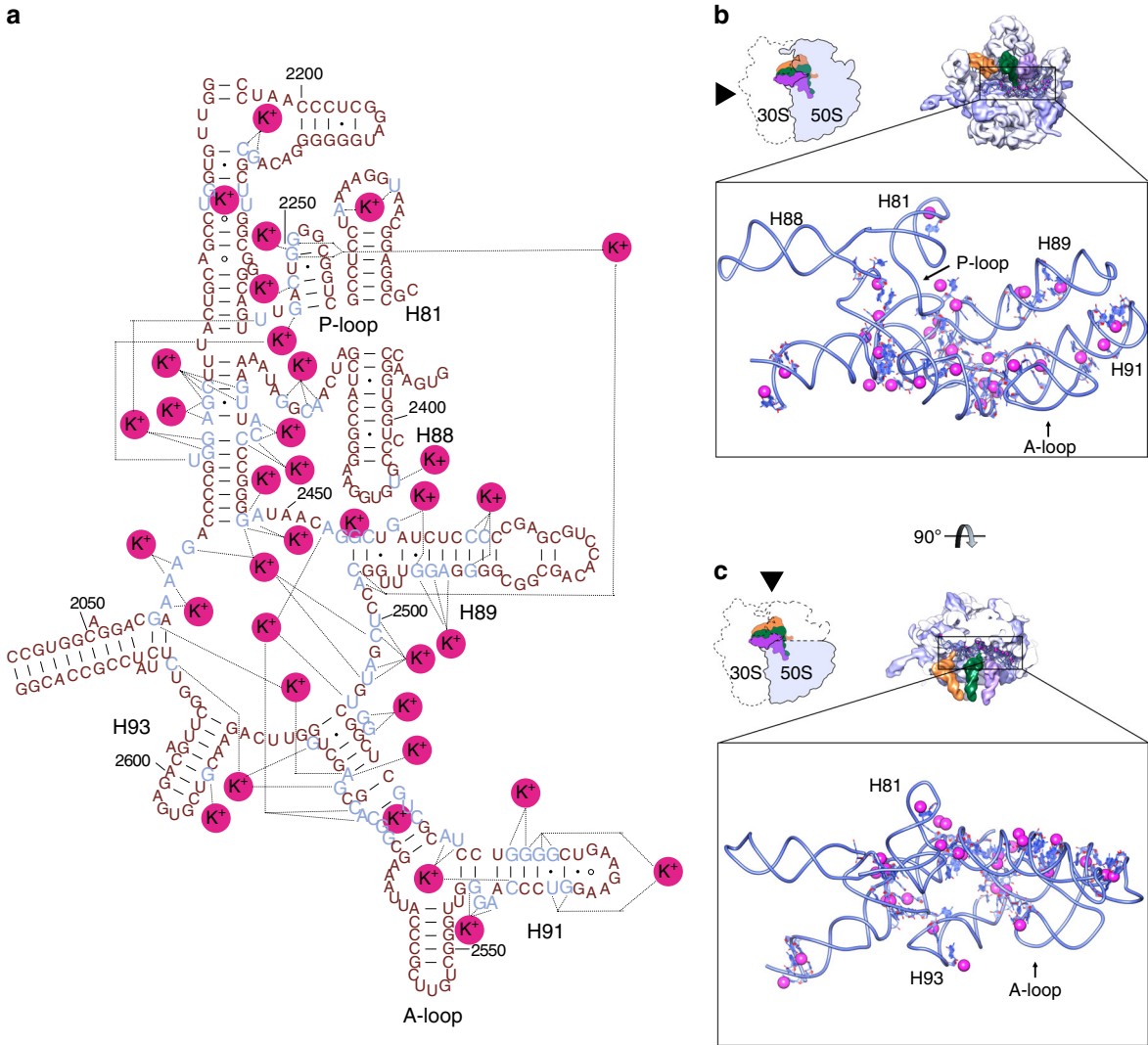

**Fig. 4** Potassium ions in the peptidyl transferase center. **a** Mapped K⁺ on secondary structure representation of PTC. The 23S rRNA secondary structure was adapted from H. Noller's lab web site (http://rna.ucsc.edu/rnacenter/images/figs/thermus_23s_2ndry.jpg link active on 04 Sept 2018). **b** Interface view of the 50S (23S, 5S in light blue, proteins in blue) with three tRNAs. In close up view only PTC and K⁺ ions (with increased van der Waals radius) are shown. **c** Interface view of the 50S, with central protuberance omitted (colors are as in **b**)

the presence of potassium has also been shown to be essential for these experiments, however, its role might have been understated due to a wider range of concentration tolerance[25].

The universal method of localization of metal ions in macromolecular structures is the analysis of ion coordination and solvent environment; albeit, this method is subject to severe limitations and does not provide unambiguous assignment even in case of atomic resolution structures[43]. The structures of large macromolecular dynamic complexes generally have poor resolution statistics; the issue is compounded by the simultaneous presence of various ions that can be either co-purified or introduced from the solvent. Taking into account the corresponding values of coordinate errors and atomic displacement parameters, except at very high resolution, it appears almost impossible to distinguish between e.g., $Mg^{2+}$, $Na^+$ or $K^+$ based only on average M…..O coordination distances (2.1, 2.4, and 2.8 Å, respectively) both by means of manual inspection or automated modelling software protocols. In addition, even at high resolution, the experimentally deduced electron densities are time-averages and, thus, it is not straightforward to assess the simultaneous presence of ions when in proximity (this can be sometimes solved by consideration of dynamics and partial occupancies, see e.g.,

ref. [70]). Very few experimental approaches allow tackling such problems. X-ray spectroscopy can provide the oxidation state and coordination of individual atoms at very high resolution[71]. However, deconvoluting the signal from several atoms in different environments is difficult and it is unable to locate the atoms within large macromolecules. Solid state nuclear magnetic resonance and scanning transmission electron microscopy showed the potential in metal ions identification, but possess various limitations[72,73]. Thus, nowadays anomalous X-ray diffraction provides the only method to unambiguously assign the nature of metal ions in macromolecular structures, especially in cases with multiple different ions (reviewed in ref. [48]). However, for light ions like $Mg^{2+}$, $Na^+$ or $K^+$ the values for the anomalous contribution to the scattering f″ are only small within the wavelength range available at typical synchrotron beamlines for macromolecular crystallography. At a wavelength of $\lambda = 2$ Å, f″ for $Mg^{2+}$ is 0.29 e⁻ and 1.68 e⁻ for $K^+$, which means a partially occupied $K^+$ could be mistaken for a $Mg^{2+}$ atom. Here, we avoid this ambiguity by measuring datasets at two wavelengths, above and below the potassium K-absorption edge. The anomalous scattering factor f″ for K varies significantly between the two wavelengths (f″$_{3.5\,keV}$ = 0.44 e⁻ vs. f″$_{3.7\,keV}$ = 3.90 e⁻), while the

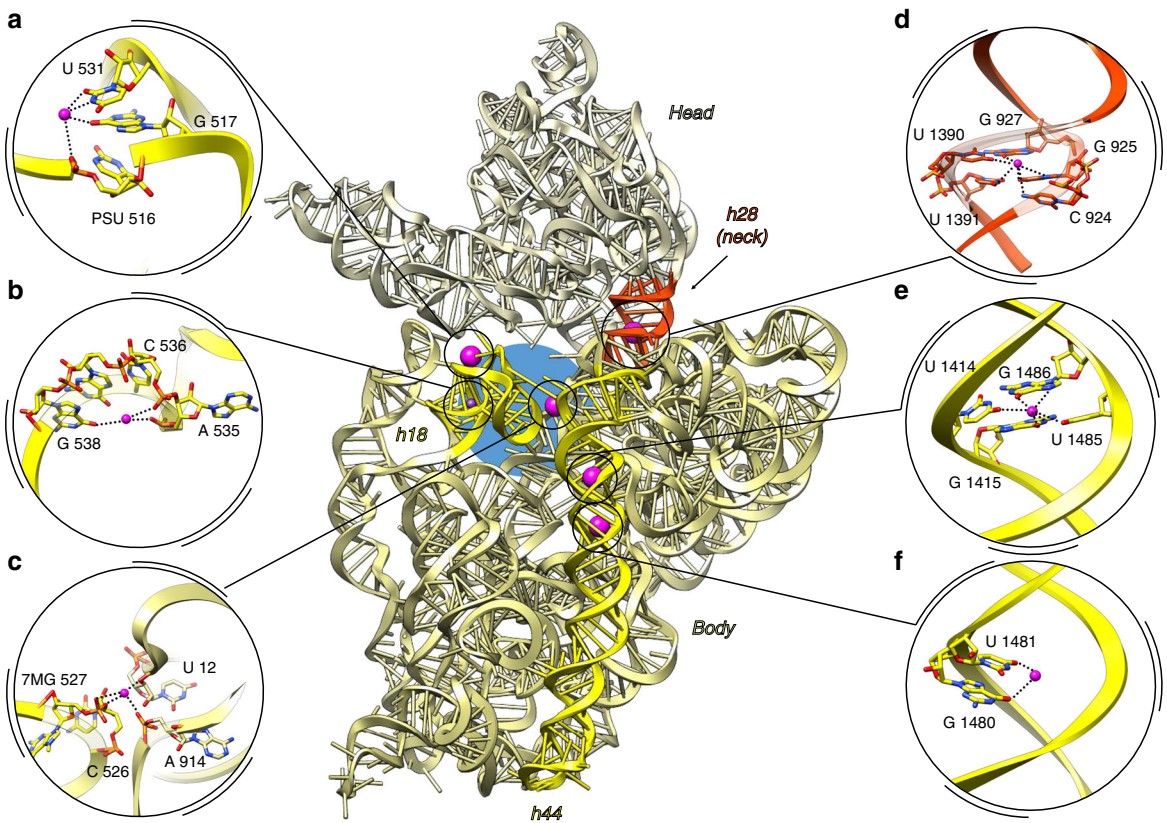

**Fig. 5** Potassium ions support essential structural elements of 30S subunit. **a–c** decoding center forming helix 18, **d** neck, **e**, **f** helix h44. 16S rRNA is presented in ribbon (center). The head is colored in khaki, body in light olive green. At the interface portion of 16S rRNA, h44 and h18 are highlighted in yellow, neck between the head and the body colored in orange-red. Decoding center is marked with a blue circle. One of the potassium ions coordinated in h18 (**b**) was found only in one ribosome of the asymmetric unit of initiation complex and it is colored in violet and represented as small sphere in the overview panel

one for Mg is almost constant ($f''_{3.5\,keV} = 0.85\ e^-$ vs. $f''_{3.7\,keV} = 0.77\ e^-$). Hence, $K^+$ cation positions can be unambiguously determined by peaks in the anomalous difference map calculated from data above the absorption edge, which are not present in the data below.

In the current paper, we present the direct experimental assignment of $K^+$ ions in the full 70S ribosome structure by long-wavelength X-ray crystallography. Registering long-wavelength diffraction from ribosome crystals became possible thanks to the novel long-wavelength MX beamline at Diamond Light Source[51]. The resolution of the datasets at such long wavelengths is limited by the strong absorption of X-rays from the crystals, the surrounding mother liquor, and sample mounts. Nevertheless, our data allowed us to unambiguously assign about 30% of the metal sites as $K^+$ and assign new ions ($K^+$) in regions that were attributed to protein component.

In this work, we elucidate the role of $K^+$ in protein synthesis at the three-dimensional level. The distribution of $K^+$ ions over the whole mass of the ribosome indicates that this ion is as important as $Mg^{2+}$. We show that potassium ions are involved in the stabilization of main functional ligands such as messenger RNA and transfer RNAs, as well as ribosomal RNAs and ribosomal proteins, via the interaction with nitrogen and oxygen atoms of side chain residues, nucleotide bases, polypeptide or sugar-phosphate backbones. These observations suggest more global and general functions of $K^+$ ions in ribosomal organization rather than its role as a stabilizer of particular regions of the ribosome or particular type of interactions. The environment of some of the identified $K^+$ ions suggested a hydrated state (Figures 3a, b), however, we cannot reliably model water molecules at 3.0–3.5 Å resolution.

Our data demonstrate that $K^+$ ions preserve ribosome integrity and the architecture of the essential functional regions. One of such regions is the decoding center, responsible for the accommodation of correct aminoacyl-tRNA (A-tRNA) in the A-site of the ribosome. Previous studies on isolated 30S subunit model identified significant structural rearrangements happening upon binding of the tRNA to the A-site (termed 'domain closure')[74]. Later, our studies of the functional complexes of the full 70S ribosome[53,58–62,75], showed that upon binding of aa-tRNA to the decoding center, the 70S ribosome undergoes only small conformational changes to proceed from the initiation to the elongation state. This movement, named 'shoulder locking'[53], displaces the 30S shoulder domain approximately 2–3 Å towards the neck, while the other parts of the 30S subunit remain immobile. It leads to a contraction of the downstream mRNA tunnel and, as a result, a network of non-specific interactions between the 16S rRNA and mRNA nucleotides forms[53], allowing tighter binding of mRNA. Our current study illustrates that two potassium ions serve as coordinators of these conformational rearrangements in the decoding center upon binding of A-tRNA.

Notably, both these ions were previously assigned as magnesium[53,55,56,58]. The ion at the third position of codon–anticodon interaction was initially observed in the structure of the 30S subunit co-crystallized with truncated mimics of mRNA and tRNAs; it was suggested to coordinate to 2′ OH of nucleotide (+6) of mRNA with the O2 of C518 (16S rRNA) and the main-chain carbonyl of Pro45 (protein uS12)[55]. Later, coordination of this ion (still assigned as $Mg^{2+}$) was re-evaluated using the crystal structure of the 70S containing long mRNA and full-length tRNA[58]. Based on careful analysis of the environment

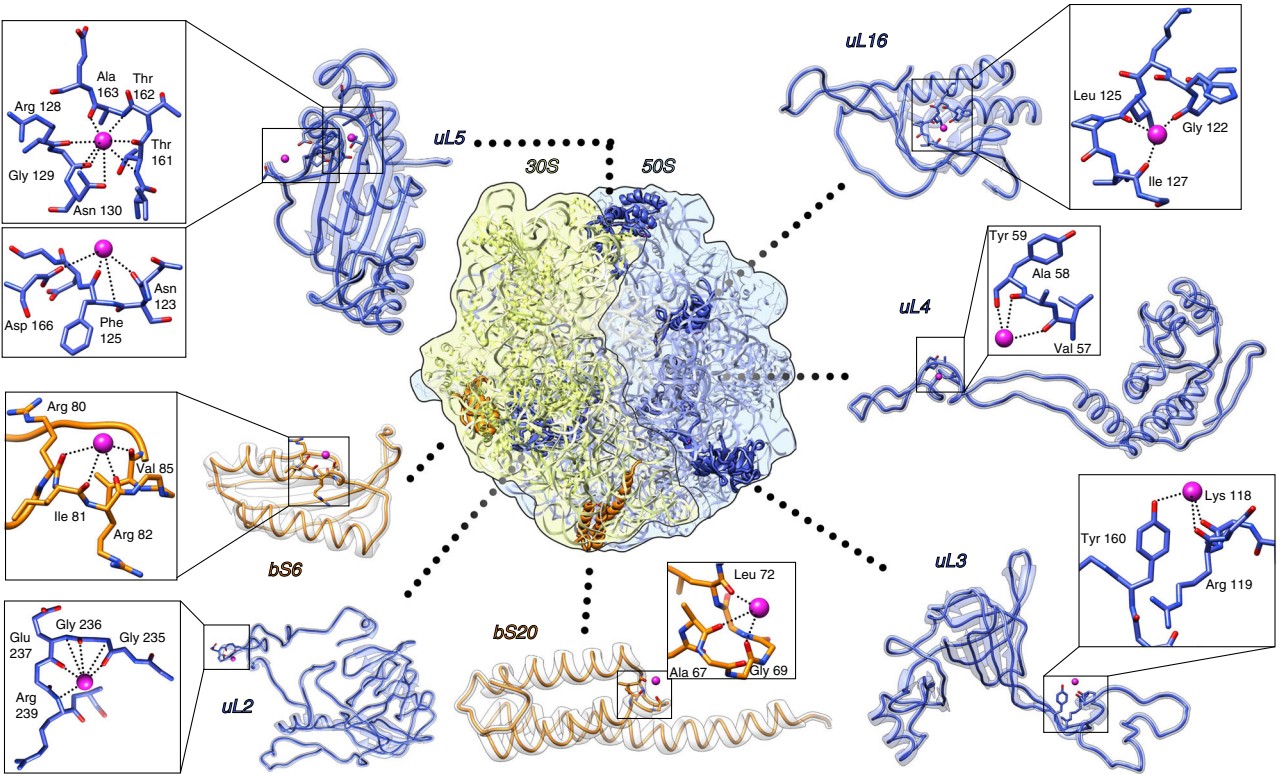

**Fig. 6** Interaction of K$^+$ ions with the ribosomal proteins from 30S subunit and from 50S subunit. K$^+$ ions are shown as magenta spheres, 30S proteins in orange, 50S proteins in blue

around the third mRNA-tRNA base pair in the A-site, the identity of these atoms was recently called into question[76,77]. The stabilization role of hydrogen bonding between 2′ OH of mRNA nucleotide at the third position with O6 of G530 of the 16S rRNA[55] remains disputable, since the distance between these two atoms varies in different crystal structures, regardless of the presence of cognate or near-cognate base pair.

The P-site tRNA has also differences in ion composition in our two complexes. In the EC, two Mg$^{2+}$ and three K$^+$ ions stabilize the P-tRNA (Supplementary Fig. 3a), while three Mg$^{2+}$ ions are found in the IC. Interestingly, in IC, two Mg$^{2+}$ stabilize the ASL of P-tRNA, and one Mg$^{2+}$ ion is coordinated to its CCA-end. In the EC, ions are mostly localized in the body of the P-tRNA. These differences, however, might be related to the nature of tRNA.

Not much is known so far about the role of ions in the mRNA path (namely the P/E and A/P kinks). In our EC structure, extra electron density was observed in both kinks. As we mentioned above, the presence of electron density in these pockets is not uniform, and the absence of metals in IC might be caused by incorrect pockets formation primarily due to the different conformation of mRNA. The binding of non-cognate E-tRNA (same tRNA$^{fMet}$ as in the P-site) in the IC is unspecific and thus may not be driving the proper formation of the P/E kink pocket. Conversely, in the EC, the poly-U mRNA forms base-pairing interactions with the cognate tRNA$^{Phe}$ in all three binding sites (including the E-site[78]) and therefore has a more controlled conformation. Additionally, the metal ion in the P/E kink appears to be coordinated to the 2-methylthio-N6-isopentenyladenosine modified nucleotide 37 in tRNA$^{Phe}$ (see Fig. 3b). The presence of the modification on nucleotide 37 of the tRNA is suggested to stabilize the first codon–anticodon base pair by stacking interactions[53] and to hinder possible translational frame-shifting[79–81]. The potassium ion coordination in the P/E-kink region might be a part of this mechanism. Regarding the A/P kink region, we

have observed additional electron density in the EC only, where mRNA is locked in its position upon the binding of A-tRNA. The absence of an "A-tRNA-lock" could be one of the reasons for the absence of the same density in the A/P kink of IC. Meantime, analysis of previous crystal structures of various functional 70S ribosomes in complex with A-/P/-E-tRNAs (with 2 ribosomes per ASU), solved by our group, showed that the density in the A/P kink region was not always present. In 33 structures we observed (and modeled as Mg$^{2+}$) difference map peaks in the A/P kink region in 35 ribosomes out of 66[53,58–62]. The four phosphate groups (16S rRNA 1401 and 1402; mRNA +4 and +5) are placed in close to planar arrangement, resulting in 3.1–3.9 Å M....O distances. The refined positions of modeled "Mg$^{2+}$" ions are spread, however, in a sphere of ~1 Å diameter. The bond distances are clearly too long for inner-sphere Mg$^{2+}$ or K$^+$ ion binding, and the absence of observed anomalous signal in this pocket in the current study suggests that the probability of the ligand being a K$^+$ ion is very low (Fig. 3c).

To date, the only study attempting experimentally to identify and distinguish monovalent and divalent ions in the ribosome structure was done using the large ribosomal subunit of archaea *Haloarcula marismortui*[82]. In that work 50S subunits were crystallized and treated in the presence of K$^+$, NH$_4^+$, Cd$^{2+}$ and excess of Na$^+$, then were soaked in excess of Rb$^+$ ions. By comparing the data from native crystals with anomalous diffraction of Rb$^+$ treated crystals authors identified and localized 82 Na$^+$ and 2 K$^+$; also, based on electron density features and geometry considerations, 166 Mg$^{2+}$, and 5 Cd$^{2+}$ ions were assigned. The two K$^+$ ions assigned in that work agree with the assignment in our structure: one K$^+$ is in the PTC (coordinating to G2061, G2447, C2501, and U2503) and the other one in H11 of the 23S rRNA (coordinating to C192, U193, A202).

Notwithstanding, K$^+$ ions can be found in many ribosome structure models deposited by different research groups. We have

analyzed and aligned all $K^+$-containing models of the ribosomes and subunits available in the Protein Data Bank (PDB). They included 67 structures of *H. marismortui* 50 S subunit, 2 structures of *Deinococcus radiodurans* 50S subunit, 6 structures of *E. coli* 70S ribosome, 32 structures of *T. thermophilus* 70S ribosome and 35 structures of *T. thermophilus* 30S subunit. Among them, PDB ID 2UUB[83] has maximum agreement of $K^+$ ions with our structures (5 out of 35). Despite the presence of $K^+$ ions in many deposited models of the ribosome, only a few papers describe how metal ions were assigned. Overall the assignment was tentative, either relying on agreement between peak selection in the $F_o - F_c$ maps and expected geometric parameters for ligand binding[82,84] or retaining the bias from structures with the highest resolution, used as initial reference during model building and refinement[85–87]. Notably, these high-resolution structures[88], that were used as templates for metal assignments in many *T. thermophilus* 30S subunit structures, contain 62 and 73 $K^+$ ions per structure, however the article lacks any information about metal ions assignment as well as experimental evidence.

As was shown for $Na^+$ in the structure of 50S ribosomal subunit of a halophilic archaea *H. marismortui*, monovalent ions have some preferential binding regions, such as RNA major grooves or stacking G–G pairs[82]. In our structures of 70S ribosomes of thermophilic bacterium *T. thermophilus* we observe similar tendencies for $K^+$ ions. It would be interesting to see how the $Mg^{2+}/K^+$ ratio and these metal ions distribution along the ribosome correlates with environmental niches of different Gram-positive and Gram-negative bacteria, and whether such correlation can be extrapolated into functional activity. On one hand, it was shown that the cellular concentrations of $K^+$ in non-stressed Gram-positive bacteria are usually much higher than their Gram-negative counterparts (for review see ref. [89]). On the other hand, protein synthesis is one of the most ancient and conserved processes in all living cells and such a difference can be related only to osmotic homeostasis rather than enzymatic processes. Additionally, a plethora of biochemical studies demonstrated the requirement of both ions for proper ribosome performance in different organisms (rev. in ref. [25]). Thus, it is plausible that the number of ions and their ratio in the ribosome will be comparable in all bacterial species. In this case, predominance of G–G coordination in *T. thermophilus* simply coincides with the abundance of G/C content in its ribosome. In conclusion, metal ions, especially alkali metal ions (i.e., $Na^+$ and especially $K^+$ as main internal cellular ions[90]) are ubiquitous components of biological systems that provide functionality for essential macromolecules, thus playing a more important role than simple ionic buffering agents or mediators of solute exchange. Moreover, $K^+$ is an important element in the structural organization of biological macromolecules. However, it is not trivial to identify $K^+$ ions in structures using conventional structure determination techniques, and it is particularly complicated for large complexes. Here, we presented the direct experimental assignment of $K^+$ ions in the structure of 70S ribosome, utilizing long-wavelength X-ray diffraction available at the unique in vacuum beamline I23, at Diamond Light Source (UK). Two functional complexes, representing two different functional states of the ribosome, demonstrated the role of $K^+$ in stabilization of ligands on the translation machinery.

Our work adds deeper insights into the mechanism of protein synthesis and open another dimension in understanding of ribosome organization. We show that some regions (e.g., the decoding center) require very precise localization, coordination and nature of metal ions. In turn, dynamical regions, such as the intersubunit bridge B2c or the A/P-kink, tolerate different ions depending on the state of the ribosome and metal-binding pocket conformation. Our observations display contrasting behaviors for the interactions of potassium and magnesium ions with ribosomal complexes. While magnesium ions tend to bind in pockets around anionic phosphate oxygen atoms with tight geometrical constraints[43], potassium ions interact with backbone carbonyl groups in protein bending folds and hydroxyl group of riboses or carbonyl groups on bases[77], especially guanine nucleotides, with a variable number of ligands and larger distance variations.

## Methods

**70S ribosome purification and crystallization.** 70 S ribosomes were purified from *Thermus thermophilus* cells, as described previously[53,91]. Uncharged native individual tRNA$^{Phe}$ and tRNA$^{fMet}$ from *E. coli* were purchased from Chemical Block (Russia). All mRNAs were purchased from Thermo Scientific (USA) and deprotected following the supplier procedure. The exact sequences were as follows: mRNA(IC) = GGCAAGGAGGUAAAAAUGA$_9$; mRNA(EC) = U$_{27}$GGCAAG-GAGGU$_{22}$. The ribosomal complexes were formed in 10 mM Tris-acetate pH 7.0, 10 mM NH$_4$Cl, 50 mM KCl, 9 mM Mg(CH$_3$COO)$_2$, at 37 °C for 20 min. For all complexes, the 70S ribosomes (3 μM) were incubated with fivefold stoichiometric excess of mRNA and three to fivefold excess of tRNA[53].

Crystals were grown at 24 °C via vapor diffusion in sitting-drop plates (CrysChem, Hampton Research). The ribosomal complex (2 μL) containing 2.8 mM Deoxy Big Chaps (CalBioChem) was mixed with the equal volume of the crystallization solution composed of 3.7–4.1% (w/v) PEG 20,000, 3.7–4.1% (w/v) PEG550mme, 100 mM Tris-acetate, pH 7.0, 100 mM KSCN. The crystals grew for 2–3 weeks and were then dehydrated by exchanging the reservoir for 60% (v/v) 2-methyl-2,4-pentanediol. Prior to plunge freezing in liquid nitrogen, crystals were cryo-protected by the addition of 30% (v/v) 2-methyl-2,4-pentanediol and 14 mM Mg(CH$_3$COO)$_2$. In order to reduce the solvent content and the absorbance of the loop, we cryo-cooled crystals in elliptical loops either made of polyimide (Litholoops$^{TM}$, Molecular Dimensions Ltd, Newmarket, UK) or laser-cut (Scitech Precision Ltd, Oxfordshire, UK) from black Kapton® B (DuPont, USA). The crystals were further transferred into the vacuum vessel using an adapted cryo-transfer system (Leica VCT100).

**Data collection, model building, and structure analysis.** The data were collected at Diamond Light Source I23 beamline[51], equipped with a Pilatus 12M (Dectris AG, Switzerland) detector, at two different wavelengths, 3.351 and 3.542 Å, using inverse beam method with 20° wedges. Data were processed using XDS[92] and half-datasets were merged using XSCALE (Supplementary Tables 2, 3). The data were collected from several crystals, however, non-isomorphism forbade using multi-crystal averaging to improve the signal and the datasets were treated separately. The structures were solved by molecular replacement using the deposited models (PDB ID 4V6G and 4V6F) with removed metal ions. From the data, omit maps were generated in Phenix[93] using models with removed metal ions. The anomalous maps from long-wavelength data were generated using ANODE[94]. The positions of anomalous peaks higher than 4.0σ as output by ANODE from datasets both above and below potassium K-edge were inspected in COOT[95] and compared with difference peaks in native omit maps. Potassium ions were modeled in the positions, where positive omit difference density peaks (> 3.5σ) overlapped with the anomalous difference peaks above the edge but not the anomalous difference peaks below the edge. Remaining positive difference omit peaks were modeled as Mg (H$_2$O)$_6^{2+}$/Mg$^{2+}$ ions. The models were then refined in Phenix against deposited diffraction data (Supplementary Table 4). We have applied a conservative cutoff of 4.0σ to anomalous map peaks to avoid false positives. The majority of peaks between 3.5 and 4.0σ corresponded to phosphorus and sulfur atoms and some may correspond to less-defined potassium sites.

The interactions of assigned $K^+$ ions with ribosome components were visualized in UCSF Chimera[96] using "Find Clashes/Contacts" function which identifies interatomic contacts based on van der Waals (VDW) radii of interacting atoms with overlap allowance of −0.5 Å. The overlap between two atoms was defined as the sum of their VDW radii minus the distance between them and minus an allowance for potentially hydrogen-bonded pairs:

$$\text{overlap}_{ij} = \text{rVDW}_i + \text{rVDW}_j - d_{ij} - \text{allowance}_{ij}$$

Default VDW values were assigned in Chimera based on the atom type. For $K^+$ ions with default coordination number (6), the radius was assigned as 1.38 Å based on the CRC Handbook of Chemistry and Physics, 82nd edition[20]. In the absence of explicit hydrogens for C, N, O, and S the software uses default VDW radii based on the ProtOr set[97]. For P, in the context of molecules (as opposed to singleton ions), atom radii 1.871 Å derived from the Amber parm99 parameters[98] [RVDW = (R*)/(2$^{1/6}$)] was used by Chimera. These selection criteria, though, do not take into account possible outer-sphere coordination, nor identify complete inner-sphere coordination due to absence of solvent molecules from the model.

The potassium ions and their environments discussed in the manuscript are summarized in Supplementary Data 1.

**Reporting summary.** Further information on research design is available in the Nature Research Reporting Summary linked to this article.

## Data availability

The data that support the findings of this study are available from the corresponding author upon reasonable request. All datasets and refined models are deposited in the Protein Data Bank under accession codes 6QNQ (IC) and 6QNR (EC).

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

## Acknowledgements

We thank the staff at Diamond Light Source for their support. We are grateful to the anonymous referees for their helpful and constructive comments. E.W. would like to thank P. Auffinger (ARN, IBMC, CNRS) for numerous discussions on ion binding to RNA. This work was supported by the French National Research Agency grants ANR-15-CE11-0021-01 (to G.Y. and E.W.), ANR-16-CE11-0007-01 (M.Y.), "La Fondation pour la Recherche Médicale" DBF20160635745, France (to G.Y.) and the Russian Government Program of Competitive Growth of Kazan Federal University (to I.K. and M.Y.) This study was also supported by the grant ANR-10-LABX-0030-INRT, a French State fund managed by the Agence Nationale de la Recherche under the frame program Investissements d'Avenir ANR-10-IDEX-0002–02.

## Author contributions

A.R., G.Y. and A.W. conceived the project; A.R. and I.K. performed data collection and analysis with assistance from K.E.O., R.D., V.M. and A.W.; A.R. and I.K. wrote the manuscript in consultation with E.W., M.Y. and A.W. All authors discussed the results and contributed to the final manuscript.

## Additional information

**Competing interests:** The authors declare no competing interests.

**Journal Peer Review Information:** *Nature Communications* thanks Alexander Serganov and the other anonymous reviewer(s) for their contribution to the peer review of this work. Peer reviewer reports are available.

