## [Peer Review File · Nature Communications]

Reviewers' comments:

Reviewer #1 (Remarks to the Author):

The authors present one of the first detailed analyses of ion coordination to the ribosome, using anomalous X-ray diffraction (long wavelength X-ray diffraction) to help localize potassium and magnesium. This paper is ground-breaking and fills a key gap in knowledge, namely, the detailed interactions of coordinated ions with the ribosome. The figures are not only clear but vivid and make the structures come alive. This is an important paper that will likely serve as a key reference in the future. The work should be published in Nature Communications as soon as possible, apart from minor revisions.

Because the ribosome is a relatively large complex, the resolution has been limited to $> 3 \text{ \AA}$ in most cases, making ion placement uncertain. The authors have produced improved positions for magnesium and potassium and discovered new important coordination geometries for the mRNA, tRNA, intersubunit bridges, and other areas.

Minor considerations:

The limitations of the methodology need to be discussed much more thoroughly. Magnesium interactions can be classified into (1) coordinated/chelated/inner sphere, (2) outer sphere, and (3) continuum. The authors only treat (1) and ignore (2) and (3) class magnesiums, which certainly play a role. They only mention: "These selection criteria, though, do not take into account possible outer-sphere coordination, nor identify complete inner-sphere coordination due to absence of solvent molecules from the model." If the authors are not able to treat (2) and (3) in their calculations, they need to at least provide a thorough discussion of these types of ion interactions (see Hayes, et al., JACS 2012). In addition, they miss a key reference regarding continuum descriptions of ions and the ribosomes (Baker, et al., PNAS 2001 – this paper has 5530 citations and is a significant oversight by the authors). Also, polyamines are known to have important effects on protein synthesis but are only mentioned in passing. These should be discussed in detail – especially the possible implications of ignoring these.

The paper would be strengthened by more analysis of the intersubunit bridges. They could add an addition figure about the bridges.

Currently, the abstract reads more like a methodology paper than a scientific paper. The impact would be improved if the authors high light the specific findings regarding the ion coordination with the mRNA and intersubunit bridges and important regions of the ribosome.

Reviewer #2 (Remarks to the Author):

Referees' report: "Anomalous magnesium": role of potassium ions in structure and function of the 70S ribosome revealed by long-wavelength X-ray diffraction" by Alexey Rozov et al.

Cations are critical for the folding of RNA, for its function, and for the assembly of large macromolecular machines that contain RNA. Physiologically, the most important cations are magnesium and potassium (although others play roles as well). In general, divalent magnesium is considered the most important cation, as its chemical properties make it well suited for interactions with RNA (especially the phosphates) and give it potent charge neutralization power to drive RNA

folding. However, it is clear that monovalent ions like potassium are also critical and can bind to RNA specifically in functionally important ways. In high resolution structure determination, the identity of a cation bound specifically to an RNA can be very ambiguous: is that blob of density a water molecule? A magnesium? Or something else? Often a magnesium is placed by default. Over the last few years, it has become clear that many structures contain incorrectly assigned ions, in part deduced by examining the coordination geometry of the assigned cation. This is certainly true in ribosome structures. However, experimental methods to unambiguously determine the identity of a bound cation are non-trivial and thus this issue is largely ignored or "swept under the rug." Unfortunately, this limits our ability to truly fundamentally understand the molecules of interest.

In this study, Rozov et al. tackle this by using long-wavelength X-ray radiation and crystallography to directly detect the presence and location of potassium ions in two 70S ribosome structures. They are able to reassign over a hundred ions as potassium, which are found in diverse parts of the ribosome and thus are almost certainly involved in processes spanning peptide bond formation, translocation, subunit association, decoding, etc.

I found this work to be of high quality, timely, creative, and well executed. The data support the conclusions, which are clearly presented. Overall, it is an important contribution and I think that the field needs to see very soon. I have only a few comments that the authors should address, which I hope will improve an already excellent study.

1. To the non-aficionado, the term "initiation complex" is confusing because of the tRNA in the E site. Although the authors attribute this to the high concentration of tRNA present, wouldn't this also place tRNA in the A site? Any chance the P or E site tRNA is carryover from the ribosome prep? Maybe a bit more explanation here.
2. I was initially confused by the fact that the statistics in table 1 and 2 show data to lower resolution, but then the refinement statistics in table 3 were to higher resolution. The methods section cleared this up, but to make thing even clearer: a) this strategy should be mentioned in the main text, and b) the accession numbers of the data used for the refinement should be mentioned and this part of the methods "fleshed out" a bit more.
3. I think a figure panel showing examples of the density that was used to assign potassium ions would be very useful. Perhaps one showing the overlay of anomalous difference with 2Fo-Fc, or something like that.
4. In figure 1c, I found the terms "re-assigned Mg²⁺" versus "assigned Mg(H₂O)₆²⁺" a bit confusing.
5. As I was reading, I kept thinking about how many sites might contain a mix of potassium or magnesium (or indeed, other ions). That is, how many are "nonspecific" cation binding sites? I suspect an occupancy refinement or analysis would be difficult and perhaps prone to artifacts, and thus it might be hard to make conclusions in this regard, but I do think it is worth a paragraph of discussion. This will prevent readers from concluding that all sites are specific for a certain cation.
6. Related to the above comment, does peak height in the anomalous map versus peak height in the 2Fo-Fc map imply anything about occupancy or the specificity of the site for potassium (is the ratio meaningful)? Do the sites with the clearest potassium coordination geometry also show more anomalous signal? I would like to see the map peak heights included in Table S1, since these data might be useful to other researchers from a crystallographic, biophysical, and methodological standpoint.

Reviewer #3 (Remarks to the Author):

The study of Yusupova and coworkers is focused on the important issue of metal identity in the ribosome structure. It has been long known that ribosomes require Mg^{2+} and K^+ cations for proper functioning; however assignment of metal cations in the ribosome structure has not been done accurately and mechanistic understanding of their roles remain poorly understood. The current study attempts to address this issue by the cutting-edge methodology presently most suited for the goal. In my opinion, this is a spectacular tour de force study that significantly advance (within experimental limits, of course) our understanding of the location of K^+ cations in the bacterial ribosome. It is most exciting to see K^+ cations located in the functionally important regions of the ribosome, suggesting that K^+ cations may have important roles in catalysis, something suspected from the biochemical literature. Undoubtedly, this work will be appreciated by the broad readership of Nature Communications.

To make the manuscript more readable by non-initiated readers, I propose to elaborate on the clever experimental setup the authors have used for assigning K^+ cations and distinguishing them from the most prevalent Mg^{2+} cations. Instead of soaking the ribosomes in Rb^+ cations that mimic K^+ cations but have stronger anomalous signal, the authors successfully exploited anomalous properties of K^+ cations directly at the peak of the anomalous scattering and off-peak, where the anomalous signal of K^+ is very small. Although anomalous signal of K^+ has been used in the past for successful identification of this cation by using shorter wave lengths (e.g., Ennifar, RNA, 2006; Serganov, Nature, 2009), it should be noted that extra biochemical efforts and structural considerations (B-factors, occupancy, coordination distances and geometry) were required to confirm cation identity. Indeed, both K^+ and Mg^{2+} cations have anomalous scattering throughout the wide range of energy. At 1.84 Å wave-length used by Serganov et al., the anomalous signal of K^+ is ~6-fold higher than scattering of Mg^{2+} (1.8 e vs 0.3 e for f''). Therefore, a K^+ cation with 17% occupancy would be indistinguishable from the 100% present Mg^{2+} cation, causing ambiguity in cation assignment based solely on anomalous scattering. In the experimental design of the current work, this shortcoming is elegantly overcome. While at the 3.351 Å wave-length both K^+ and Mg^{2+} have anomalous signal (~4.0 and 0.8 e, respectively), at 3.542 Å, K^+ cation has strongly reduced anomalous scattering (0.5 e) while Mg^{2+} retains almost the same signal (~0.9 e). Therefore, significant change in anomalous scattering at the two wave length clearly indicates the presence of a K^+ cation.

Other inquires:

This reviewer thanks the authors for providing coordinates & maps for assessment. Analysis of the data showed that the authors should make some changes in the text of the manuscript as well as revisit cation assignment. Some (not all) problems are summarized in the table below.

- 1) Phosphorus atom has ~2 e anomalous signal at the wave lengths used for data collection. Some anomalous peaks coincide with Ps of the RNA and not with adjacent cations. It is believed that, despite some differences in the intensity of anomalous signal at the two wave lengths, several peaks assigned to K^+ are in fact P atoms.
- 2) In a number of instances, K^+ cations are located in close proximity to other cations, typically Mg^{2+} . It looks like cations were used to fill the density map. There are instances when two cations are located at a distance smaller than the sum of the cation radii (2.23 Å). This is a physically impossible arrangement. Given repelling charges of cations, it is also highly unlikely that cations would be positioned closely (<3 Å) to each other in many other locations. In such instances, Mg^{2+} cations must be removed. It could be very difficult to formalize criteria for removing cations since a "minimal distance" may not be the best criterion here because cations can come closely to each other if surroundings have a high density of the negative charge. The authors have to critically analyze their cation

assignments and use common sense to remove a number of cations. Residual map, if present, would correspond to the hydration sphere of K⁺. Alternatively, it could be the same K⁺ cation positioned in two close sites with partial occupancy. This reviewer has only verified K⁺ cations and it is suspected that many Mg²⁺ cation sites have the same problem.

- 3) In several cases, anomalous peaks assigned to K⁺ cations are visible only at <4 sigma levels, the levels lower than the threshold mentioned in the manuscript. The true threshold (or other considerations used for assignment) should be mentioned in the manuscript.
- 4) Page 6, line 153. How many K⁺ cations are common to both ribosomes of the asymmetric unit? Are there K⁺ cations which appear to be K⁺s in one ribosome and not in another one?
- 5) Page 6, line 155. How many K⁺ cations are common to initiating and elongating ribosomes? Was an effort made to correlate K⁺ cations between these different structures? Supplementary Movie 1 is great but it illustrates well K⁺s in the important regions of the ribosome. See the next comment.
- 6) Page 6, line 163. It could be helpful to have Supplementary Table 1 for both initiating and elongating ribosomes (by the way, what structure is STable 1 based on?).
- 7) Fig. 2, page 7. The evidence for K⁺ cations in the decoding center is convincing. Are there any biochemical data in literature to highlight this important finding?

Atom	K cation #	Data set	Chain	Comment
	5	Initiation	T	No 2Fo-Fc
	6			Too close (3 Å) to 81K. Single anomalous peak for two K ⁺ .
	9		T	Signal belongs to neighboring 11KV
Op2/469 G/1H				Missing cations; both anomalous and omit 2Fo-Fc present.
	17		T	No anomalous signal.
	43		T	Single peak for two cations. Too close (3.2 Å) to 51KV
	52		T	Anomalous peak at >3 Å distance, possibly P atom.
	53		T	No anomalous and 2Fo-Fc maps.
	54		T	Too close (2.2 Å) to 72MgX. This Mg cation is likely K ⁺ , while 54K does not exist.
	55			Too close (2.77 Å) to 103MgX. This Mg ²⁺ likely does not exist.
	57		T	Most likely, neighboring 408MgX is a K ⁺ cation according to the anomalous peak while 57K is a Mg ²⁺ cation
	59		T	Too close (2.73 Å) to 453MgX. This Mg ²⁺ likely does not exist.
A1698 and U766	62		T	Position of this cation does not match anomalous signal visible between two
				phosphates of the RNA. Most likely, neighboring 415MgX does not exist.
	65		T	Anomalous peak is positioned closer to 153MgX than to 65K.

	66		T	No 2Fo-Fc for this cation. The K ⁺ cation should be moved into the density partially occupied by 261MgX while Mg cation should be removed.
	71		T	Anomalous peak for this cation is at ~3.7 sigma level, not 4.0.
	72		T	Anomalous peak for this cation is at ~3.4 sigma level, not 4.0. Too close (2.62 Å) to 435MgX. This Mg cation should be removed.
	74		T	This cation does not have its own anomalous peak. The peak belongs to the adjacent 49KV.
	75		T	This site is strange. There are 6 cations in close proximity, including 2 K ⁺ cations. However, the anomalous signal corresponds only to 81KV while 75KT does not have the peak. Most likely, 75KT and 462MgX should be removed while hydrated 460MgX should be positioned centrally.
	80		T	Too close (2.93 Å) to 53MgX. This Mg cation should be removed.
	84, 85, 86		T	Anomalous peak at <4.0 sigma level.
	8		V	Too close (2.23 Å) to 7MgX. This Mg cation should be removed.
	24		V	Located far from anomalous peak
	36		V	Too close (2.57 Å) to 173MgX. This Mg cation should be removed.
	76		V	Too close (2.35 Å) to 331MgX. This Mg cation should be removed.
	83		V	No anomalous peak here.
	89		V	Too close (2.66 Å) to 87MgX. This Mg cation should be removed.
	3		U	No 2Fo-Fc for this cation. Most likely, adjacent 166MgY is the K ⁺ cation.
	18		U	Too close (2.15 Å) to 31MgY. This Mg cation should be removed.
	19			Too close (2.44 Å) to 340MgY. This Mg cation should be removed.
99MgY				This is K ⁺ .
	20		U	Too close (1.96 Å) to 13MgY. This Mg cation should be removed.
	23		U	Too close (2.94 Å) to 382MgY. This Mg cation should be removed.
	24		U	This cation should be moved towards the center of Fo-Fc map.
	27		U	Adjacent 39MgY is likely K ⁺ while 27K is coordinated water.
	32, 33		U	There is one anomalous peak for 1 K ⁺ , not two.

	3		W	Too close (3.63 Å) to 46MgY. This Mg cation should be removed.
	36		W	Should be positioned better in 2Fo-Fc map.
	37		W	Too close (2.25 Å) to 168MgY. This Mg cation should be removed.
	75		w	This is Mg ²⁺ cation while adjacent 169MgY is a K ⁺ cation according to anomalous peak.
	77		w	This is Mg ²⁺ cation while adjacent 234MgY is a K ⁺ cation according to anomalous peak.
		elongation		
	34		T	Too close to 603MgX and 620MgX. These two Mg cations should be removed. I think 34K is a Mg ²⁺ cation while 601MgX is a K ⁺ because it is located closer to the anomalous peak.
	35		T	496MgX should be removed.
	37		T	Too close to 75MgX. This Mg cation should be removed.
	43		T	This K ⁺ likely does not exist. It is located between two K ⁺ cations and practically has no 2Fo-Fc map.
	50		T	There are 3 cations in the same density. It is clearly 1 hydrated K ⁺ .
	54		T	54K and 76K are likely a single K ⁺ .
	55		T	Too close to 49MgX. This Mg cation should be removed.
	57		T	57KT and 117KV are probably a single hydrated K ⁺ .
	58		T	58KT and 101KV are a single hydrated K ⁺ .
	60		T	60KT and 78KV are a single hydrated K ⁺ .
	61		T	Remove adjacent 61MgX.
	62		T	Shape of the 2Fo-Fc and anomalous density maps suggests a single hydrated K ⁺ . There are 4 cations here, with distances between K ⁺ s ~3.0 Å. I do not think it is a possible arrangement.
	66		T	Also too many cations located at short distances from each other. Impossible arrangement.
	70		T	No anomalous density here.
	71		T	Too close to 153MgX.
	72		T	Too close to 156MgX.
	74		T	No anomalous density here. Too close to two Mg ²⁺ cations.
	81		T	81KT and 82KT are a single K ⁺ cation.
	86		T	86KT and 87KT are a single K ⁺ cation according to a single anomalous peak.
	90		T	There are too many closely positioned cations. Unlikely scenario.

	92		T	Here adjacent 571MgX is probably a K ⁺ while 92KT is a Mg ²⁺ according to the anomalous peak position.
	93		T	93K, 94K an 508MgX is probably a single hydrated K ⁺ .
	95		T	Too close to 265MgX. mg cation should be removed.
	96		T	This is not K ⁺ .
	101		T	This is not K ⁺ .
	34		V	Two Mg ²⁺ cations in close proximity.
	41		V	Close to 9MgX.
	42		V	Close to 307MgX.
	54		V	Close to 212MgX.
	58		V	Close to 194 Mgx.
	76		V	Probably a single K ⁺ cation instead of two here.
	78		V	Probably a single K ⁺ cation instead of two here.
	131		V	Close to 262MgX.
	133		V	There are 5 cations here while the shape of the density suggests only 3 cations.
	138		V	No anomalous signal here. 4 cations here are probably a single hydrated Mg ²⁺ cation.
	139		V	4 cations here are probably a single hydrated K ⁺ .
	140		V	Too close to 297MgX.
	148		V	A whole bunch of cations here. 148KT and 100KT are probably a single K ⁺ .

We thank all the reviewers for taking time to thoroughly analyze our manuscript and the problem we addressed. We are grateful for the constructive and helpful feedback and we strove to accommodate the suggestions wherever possible.

Reviewers' comments:

Reviewer #1 (Remarks to the Author):

The authors present one of the first detailed analyses of ion coordination to the ribosome, using anomalous X-ray diffraction (long wavelength X-ray diffraction) to help localize potassium and magnesium. This paper is ground-breaking and fills a key gap in knowledge, namely, the detailed interactions of coordinated ions with the ribosome. The figures are not only clear but vivid and make the structures come alive. This is an important paper that will likely serve as a key reference in the future. The work should be published in Nature Communications as soon as possible, apart from minor revisions.

Because the ribosome is a relatively large complex, the resolution has been limited to $> 3 \text{ \AA}$ in most cases, making ion placement uncertain. The authors have produced improved positions for magnesium and potassium and discovered new important coordination geometries for the mRNA, tRNA, intersubunit bridges, and other areas.

Minor considerations:

The limitations of the methodology need to be discussed much more thoroughly. Magnesium interactions can be classified into (1) coordinated/chelated/inner sphere, (2) outer sphere, and (3) continuum. The authors only treat (1) and ignore (2) and (3) class magnesiums, which certainly play a role. They only mention: "These selection criteria, though, do not take into account possible outer-sphere coordination, nor identify complete inner-sphere coordination due to absence of solvent molecules from the model." If the authors are not able to treat (2) and (3) in their calculations, they need to at least provide a thorough discussion of these types of ion interactions (see Hayes, et al., JACS 2012). In addition, they miss a key reference regarding continuum descriptions of ions and the ribosomes (Baker, et al., PNAS 2001 – this paper has 5530 citations and is a significant oversight by the authors).

We would like to point out that the sentence the referee is citing is related specifically to description/visualization of contacts between modelled ions and macromolecules. The reviewer is right to point out that our experimental data cannot provide information regarding the "continuum of ions" because of the lack of experimental electronic density for ions in the continuum due to the dynamics of such ions. We felt that focusing our manuscript primarily on the interpretation of our experimental results, with, as other reviewers requested, a more thorough discussion on the limits of our experimental approach and final models, would be preferable over theoretical calculations. In the same vein we cannot provide a comprehensive theoretical discussion of complete ionic atmosphere of ribosome in relation to the map of electrostatic potential. Our results provide new experimental data on the presence and positions of ions (and more specifically potassium) and hopefully our results will provide important data for scientists who specialize in the topic of theoretical electrostatic calculations.

Also, polyamines are known to have important effects on protein synthesis but are only mentioned in passing. These should be discussed in detail – especially the possible implications of ignoring these.

In the introduction section, we have added a discussion on how polyamines influence protein synthesis and their relations with Mg^{2+} ions.

The paper would be strengthened by more analysis of the intersubunit bridges. They could add an addition figure about the bridges.

We thank the reviewer for raising this important point. We have included analysis of intersubunit bridges into results and discussion. Figure 1C was updated to show intersubunit bridges and coordinated ions.

Currently, the abstract reads more like a methodology paper than a scientific paper. The impact would be improved if the authors highlight the specific findings regarding the ion coordination with the mRNA and intersubunit bridges and important regions of the ribosome.

In the manuscript abstract we aimed to highlight both the novelty of the method we utilized and the findings it allowed us to present. Unfortunately, the abstract is also subject to space constraints that we have already exceeded.

Reviewer #2 (Remarks to the Author):

Referees' report: "Anomalous magnesium": role of potassium ions in structure and function of the 70S ribosome revealed by long-wavelength X-ray diffraction" by Alexey Rozov et al.

Cations are critical for the folding of RNA, for its function, and for the assembly of large macromolecular machines that contain RNA. Physiologically, the most important cations are magnesium and potassium (although others play roles as well). In general, divalent magnesium is considered the most important cation, as its chemical properties make it well suited for interactions with RNA (especially the phosphates) and give it potent charge neutralization power to drive RNA folding. However, it is clear that monovalent ions like potassium are also critical and can bind to RNA specifically in functionally important ways. In high resolution structure determination, the identity of a cation bound specifically to an RNA can be very ambiguous: is that blob of density a water molecule? A magnesium? Or something else? Often a magnesium is placed by default. Over the last few years, it has become clear that many structures contain incorrectly assigned ions, in part deduced by examining the coordination geometry of the assigned cation. This is certainly true in ribosome structures. However, experimental methods to unambiguously determine the identity of a bound cation are non-trivial and thus this issue is largely ignored or "swept under the rug." Unfortunately, this limits our ability to truly fundamentally understand the molecules of interest.

In this study, Rozov et al. tackle this by using long-wavelength X-ray radiation and crystallography to directly detect the presence and location of potassium ions in two 70S ribosome structures. They are able to reassign over a hundred ions as potassium, which are found in diverse parts of the ribosome and thus are almost certainly involved in processes spanning peptide bond formation, translocation, subunit association, decoding, etc.

I found this work to be of high quality, timely, creative, and well executed. The data support the conclusions, which are clearly presented. Overall, it is an important contribution and I think that the field needs to see very soon. I have only a few comments that the authors should address, which I hope will improve an already excellent study.

1. To the non-aficionado, the term "initiation complex" is confusing because of the tRNA in the E site. Although the authors attribute this to the high concentration of tRNA present, wouldn't this also place tRNA in the A site? Any chance the P or E site tRNA is carryover from the ribosome prep? Maybe a bit more explanation here.

The affinity of aminoacylated/deacylated tRNAs to various sites were thoroughly studied biochemically for ~40 years (groups of Nierhaus, Wintermeyer, Kirillov). The conclusions (rev. in Graifer, *IntJMolSci*, 2015), in brief, are that tRNAs have highest affinity to the P-site, then A-site and last, E-site. Additionally P- and A- site binding is strongly influenced by the codon present in the site, with non-cognate codons precluding binding. E-site tRNA binding is not affected by the nature of the codon or its presence at all. In our structures, in the initiation complex, the codon AAA in the A-site is non-cognate for the initiator tRNA^{fMet} (specific to AUG codon) supplied for complex formation and, thus the A-site stays vacant. The approach of presenting the non-cognate codons in the A-site is routinely used to generate complexes of 70S ribosomes with vacant A-sites (Yusupov, *Science*, 2001; Jenner NSMB, 2010; Rozov *Nat Comms*, 2015).

Regarding possible co-purified contaminants, we are also conducting experiments with vacant 70S ribosomes (without supplied mRNA/tRNA) and in these structures we do not observe the presence of any detectable contaminant.

2. I was initially confused by the fact that the statistics in table 1 and 2 show data to lower resolution, but then the refinement statistics in table 3 were to higher resolution. The methods section cleared this up, but to make thing even clearer: a) this strategy should be mentioned in the main text, and b) the accession numbers of the data used for the refinement should be mentioned and this part of the methods “fleshed out” a bit more.

We understand the possible confusion about these point as our strategy was unusual. We expanded the panel for data collection and ions re-assignment in Figure 1B. We also added accession numbers of structures/data we used to the Table 3 for clarity. The experimental strategy is also described both in Results and Methods section with a specific mention of the accession numbers for the previous structures/data.

3. I think a figure panel showing examples of the density that was used to assign potassium ions would be very useful. Perhaps one showing the overlay of anomalous difference with 2Fo-Fc, or something like that.

For the cation assignments in this manuscript we have used overlays of 6-9 maps simultaneously, and assigned ~650 potassium ions in two structures. Our attempts to summarize it even in a simplified figure led us to the conclusion that such representation would not convey much information to the reader, since the figure panel would have to be restrained to 1-2 maps and 1-2 ions for readability.

4. In figure 1c, I found the terms “re-assigned Mg²⁺” versus “assigned Mg(H₂O)₆²⁺” a bit confusing.

It was indeed confusing. We left “assigned” for both cases.

5. As I was reading, I kept thinking about how many sites might contain a mix of potassium or magnesium (or indeed, other ions). That is, how many are “nonspecific” cation binding sites? I suspect an occupancy refinement or analysis would be difficult and perhaps prone to artifacts, and thus it might be hard to make conclusions in this regard, but I do think it is worth a paragraph of discussion. This will prevent readers from concluding that all sites are specific for a certain cation.

This is an important issue and we are grateful the referee for raising it. First and foremost, indeed, occupancy analysis will not be informative due to our experimental setup with data coming from different crystals, not to mention the limitations imposed by data resolution. Second, with regards to the issue of “mixed” sites, we have to introduce a classification, similar to the one used to describe the ionic atmosphere of a macromolecule (also mentioned by reviewer #1). In brief, the ions present in the solvent fraction of the macromolecular structure can be subdivided into three classes: i) directly coordinated (where atoms of the macromolecule intrude into the solvation shell of an ion); ii) coordinated via solvent molecules; and iii) continuum, which due to stochasticity of distribution cannot be detected/described by structural methods. As discussed in the text

(and answered to reviewer 1), we cannot identify the ions in the continuum because they are mobile and when flash-freezing the crystals they will occupy random positions in the matrix leading to absence of electron density. The second of the types could be partially visualized in X-ray structures if the ions are present in roughly same position in sufficient number of asymmetric units and due to weaker binding/geometry constraints can, indeed, be “mixed” sites. However, uncertainty of position/occupancy determination will make it impossible to judge the ratios or whether there is a mixture. The limitation in resolution prevents using geometrical criteria for definite identification. The first type, the directly coordinated ions have to be subdivided into two subtypes. There are sites where the macromolecules occupy multiple chelating positions in the ions’ solvation shells (the decoding center ions are a good example) and in such cases present unambiguously differentiation between Mg^{2+} and K^+ . Indeed, the geometries of these binding pockets are drastically different for these ions and, therefore, these sites cannot accommodate different ions. Another subtype will be ions which have just a single contact with the macromolecule in their immediate solvation shell. In these cases there is a possibility of ion “overlay” with according displacement, but in the limits of our experimental setup we cannot identify such cases.

To summarize, in our structures we have identified as potassium ions the sites where we have observed both anomalous and non-anomalous difference. These observations are subject to limitations of the data collection and data quality. Within these limitations we can safely conclude that we have identified some of the potassium ions bound to the ribosome (we cannot exclude that we did not identify all of these due to low anomalous signal) and some of these sites are exclusive for potassium ions.

6. Related to the above comment, does peak height in the anomalous map versus peak height in the 2Fo-Fc map imply anything about occupancy or the specificity of the site for potassium (is the ratio meaningful)? Do the sites with the clearest potassium coordination geometry also show more anomalous signal? I would like to see the map peak heights included in Table S1, since these data might be useful to other researchers from a crystallographic, biophysical, and methodological standpoint.

First, our peaks are derived from different crystals, so the heights and ratios will not be something constant and meaningful, unfortunately. Another issue we see with regards to peak heights as occupancy or specificity measures is that our structures have variable flexibility, with some fluctuations of average ADPs between different regions. These fluctuations will skew the peak heights inside one structure without reflection of K^+ occupancy or site specificity.

Reviewer #3 (Remarks to the Author):

The study of Yusupova and coworkers is focused on the important issue of metal identity in the ribosome structure. It has been long known that ribosomes require Mg²⁺ and K⁺ cations for proper functioning; however assignment of metal cations in the ribosome structure has not been done accurately and mechanistic understanding of their roles remain poorly understood. The current study attempts to address this issue by the cutting-edge methodology presently most suited for the goal. In my opinion, this is a spectacular tour de force study that significantly advance (within experimental limits, of course) our understanding of the location of K⁺ cations in the bacterial ribosome. It is most exciting to see K⁺ cations located in the functionally important regions of the ribosome, suggesting that K⁺ cations may have important roles in catalysis, something suspected from the biochemical literature. Undoubtedly, this work will be appreciated by the broad readership of Nature Communications.

To make the manuscript more readable by non-initiated readers, I propose to elaborate on the clever experimental setup the authors have used for assigning K⁺ cations and distinguishing them from the most prevalent Mg²⁺ cations. Instead of soaking the ribosomes in Rb⁺ cations that mimic K⁺ cations but have stronger anomalous signal, the authors successfully exploited anomalous properties of K⁺ cations directly at the peak of the anomalous scattering and off-peak, where the anomalous signal of K⁺ is very small. Although anomalous signal of K⁺ has been used in the past for successful identification of this cation by using shorter wave lengths (e.g., Ennifar, RNA, 2006; Serganov, Nature, 2009), it should be noted that extra biochemical efforts and structural considerations (B-factors, occupancy, coordination distances and geometry) were required to confirm cation identity. Indeed, both K⁺ and Mg²⁺ cations have anomalous scattering throughout the wide range of energy. At 1.84 Å wave-length used by Serganov et al., the anomalous signal of K⁺ is ~6-fold higher than scattering of Mg²⁺ (1.8 e vs 0.3 e for f''). Therefore, a K⁺ cation with 17% occupancy would be indistinguishable from the 100% present Mg²⁺ cation, causing ambiguity in cation assignment based solely on anomalous scattering. In the experimental design of the current work, this shortcoming is elegantly overcome. While at the 3.351 Å wave-length both K⁺ and Mg²⁺ have anomalous signal (~4.0 and 0.8 e, respectively), at 3.542 Å, K⁺ cation has strongly reduced anomalous scattering (0.5 e) while Mg²⁺ retains almost the same signal (~0.9 e). Therefore, significant change in anomalous scattering at the two wavelengths clearly indicates the presence of a K⁺ cation.

We thank reviewer for the input and useful comments. We have modified the text according to received suggestions.

Other inquires:

This reviewer thanks the authors for providing coordinates & maps for assessment. Analysis of the data showed that the authors should make some changes in the text of the manuscript as well as revisit cation assignment. Some (not all) problems are summarized in the table below.

- 1) Phosphorus atom has ~2 e anomalous signal at the wave lengths used for data collection. Some anomalous peaks coincide with Ps of the RNA and not with adjacent cations. It is believed that, despite some differences in the intensity of anomalous signal at the two wave lengths, several peaks assigned to K⁺ are in fact P atoms.

There is such a possibility, especially in cases of several adjacent PO4 groups. However, presence of F_o-F_c map and 3-4 adjacent phosphates allow us to additionally analyze coordination geometry to confirm the ion presence/identity.

- 2) In a number of instances, K+ cations are located in close proximity to other cations, typically Mg²⁺. It looks like cations were used to fill the density map. There are instances when two cations are located at a distance smaller than the sum of the cation radii (2.23 Å). This is a physically impossible arrangement. Given repelling charges of cations, it is also highly unlikely that cations would be positioned closely (<3 Å) to each other in many other locations. In such instances, Mg²⁺ cations must be removed. It could be very difficult to formalize criteria for removing cations since a “minimal distance” may not be the best criterion here because cations can come closely to each other if surroundings have a high density of the negative charge. The authors have to critically analyze their cation assignments and use common sense to remove a number of cations. Residual map, if present, would correspond to the hydration sphere of K+. Alternatively, it could be the same K+ cation positioned in two close sites with partial occupancy. This reviewer has only verified K+ cations and it is suspected that many Mg²⁺ cation sites have the same problem.

We have revisited and carefully reassessed K⁺ and Mg²⁺ assignment. We are very grateful to the reviewer for thorough analysis of the data and models. All of the issues the reviewer presents in the table (and some more) were verified and addressed. We would like to point out that the electron densities obtained by crystallography are time-averages (so we identify binding sites) and thus one cannot easily assess whether the close-by ions are simultaneously present in all the unit cells of the crystals (in other words, only one of the two close-by ions is always occupying its site, in proportions that are difficult to evaluate). Variable occupancies would require higher resolution.

- 3) In several cases, anomalous peaks assigned to K+ cations are visible only at <4 sigma levels, the levels lower than the threshold mentioned in the manuscript. The true threshold (or other considerations used for assignment) should be mentioned in the manuscript.

These occasions were reanalyzed and corrected. We maintain that the threshold is 4.0 sigma level, however, some ions were initially placed in lower signal density if they coincided with stronger signal in another ribosome of the asymmetric unit. We have added clarification to the Methods section.

- 4) Page 6, line 153. How many K+ cations are common to both ribosomes of the asymmetric unit? Are there K+ cations which appear to be K+s in one ribosome and not in another one?

We have performed pairwise comparison between both ribosomes of asymmetric unit in both complexes. Common ions are now quantified in Supplementary table 1.

- 5) Page 6, line 155. How many K+ cations are common to initiating and elongating ribosomes? Was an effort made to correlate K+ cations between these different structures? Supplementary Movie 1 is great but it illustrates well K+s in the important regions of the ribosome. See the next comment.

Please, see the answer above.

- 6) Page 6, line 163. It could be helpful to have Supplementary Table 1 for both initiating and elongating ribosomes (by the way, what structure is STable 1 based on?).

We have updated this Supplementary table (Now it is supplementary table S2). We have put additional column where every ion discussed in the table is now affiliated with ribosomes where it is present. The coordination is mainly given for the best resolved ribosome of the elongation complex (referred to as EC-A in the table). Whenever EC-A was not applicable, ion coordination was given for initiation complex (referred to as IC-B in the table). When both weren't applicable EC-B and IC-A were used as references.

- 7) Fig. 2, page 7. The evidence for K⁺ cations in the decoding center is convincing. Are there any biochemical data in literature to highlight this important finding?

Not to our knowledge. The biochemical studies we cite and many others touch upon general ionic conditions of ribosome environment, i.e. salt concentration, nature etc. In fact, without prior structural information, it seems next to impossible to track individual ions in large complexes such as ribosome due to the amounts of ions bound.

Atom	K cation #	Data set	Chain	Comment
	5	Initiation	T	No 2Fo-Fc
	6			Too close (3 Å) to 81K. Single anomalous peak for two K+.
	9		T	Signal belongs to neighboring 11KV
Op2/469 G/1H				Missing cations; both anomalous and omit 2FoFc present.
	17		T	No anomalous signal.
	43		T	Single peak for two cations. Too close (3.2 Å) to 51KV
	52		T	Anomalous peak at >3 Å distance, possibly P atom.
	53		T	No anomalous and 2Fo-Fc maps.
	54		T	Too close (2.2 Å) to 72MgX. This Mg cation is likely K+, while 54K does not exist.
	55			Too close (2.77 Å) to 103MgX. This Mg2+ likely does not exist.
	57		T	Most likely, neighboring 408MgX is a K+ cation according to the anomalous peak while 57K is a Mg2+ cation
	59		T	Too close (2.73 Å) to 453MgX. This Mg2+ likely does not exist.
A1698 and U766	62		T	Position of this cation does not match anomalous signal visible between two phosphates of the RNA. Most likely, neighboring 415MgX does not exist.
	65		T	Anomalous peak is positioned closer to 153MgX than to 65K.
	66		T	No 2Fo-Fc for this cation. The K+ cation should be moved into the density partially occupied by 261MgX while Mg cation should be removed.
	71		T	Anomalous peak for this cation is at ~3.7 sigma level, not 4.0.
	72		T	Anomalous peak for this cation is at ~3.4 sigma level, not 4.0. Too close (2.62 Å) to 435MgX. This Mg cation should be removed.
	74		T	This cation does not have its own anomalous peak. The peak belongs to the adjacent 49KV.

	75		T	This site is strange. There are 6 cations in close proximity, including 2 K ⁺ cations. However, the anomalous signal corresponds only to 81KV while 75KT does not have the peak. Most likely, 75KT and 462MgX should be removed while hydrated 460MgX should be positioned centrally.
	80		T	Too close (2.93 Å) to 53MgX. This Mg cation should be removed.
	84, 85, 86		T	Anomalous peak at <4.0 sigma level.
	8		V	Too close (2.23 Å) to 7MgX. This Mg cation should be removed.
	24		V	Located far from anomalous peak
	36		V	Too close (2.57 Å) to 173MgX. This Mg cation should be removed.
	76		V	Too close (2.35 Å) to 331MgX. This Mg cation should be removed.
	83		V	No anomalous peak here.
	89		V	Too close (2.66 Å) to 87MgX. This Mg cation should be removed.
	3		U	No 2Fo-Fc for this cation. Most likely, adjacent 166MgY is the K ⁺ cation.
	18		U	Too close (2.15 Å) to 31MgY. This Mg cation should be removed.
	19			Too close (2.44 Å) to 340MgY. This Mg cation should be removed.
99MgY				This is K ⁺ .
	20		U	Too close (1.96 Å) to 13MgY. This Mg cation should be removed.
	23		U	Too close (2.94 Å) to 382MgY. This Mg cation should be removed.
	24		U	This cation should be moved towards the center of Fo-Fc map.
	27		U	Adjacent 39MgY is likely K ⁺ while 27K is coordinated water.
	32, 33		U	There is one anomalous peak for 1 K ⁺ , not two.
	3		W	Too close (3.63 Å) to 46MgY. This Mg cation should be removed.
	36		W	Should be positioned better in 2Fo-Fc map.
	37		W	Too close (2.25 Å) to 168MgY. This Mg cation should be removed.
	75		w	This is Mg ²⁺ cation while adjacent 169MgY is a K ⁺ cation according to anomalous peak.

	77		w	This is Mg ²⁺ cation while adjacent 234MgY is a K ⁺ cation according to anomalous peak.
		elongation		
	34		T	Too close to 603MgX and 620MgX. These two Mg cations should be removed. I think 34K is a Mg ²⁺ cation while 601MgX is a K ⁺ because it is located closer to the anomalous peak.
	35		T	496MgX should be removed.
	37		T	Too close to 75MgX. This Mg cation should be removed.
	43		T	This K ⁺ likely does not exist. It is located between two K ⁺ cations and practically has no 2Fo-Fc map.
	50		T	There are 3 cations in the same density. It is clearly 1 hydrated K ⁺ .
	54		T	54K and 76K are likely a single K ⁺ .
	55		T	Too close to 49MgX. This Mg cation should be removed.
	57		T	57KT and 117KV are probably a single hydrated K ⁺ .
	58		T	58KT and 101KV are a single hydrated K ⁺ .
	60		T	60KT and 78KV are a single hydrated K ⁺ .
	61		T	Remove adjacent 61MgX.
	62		T	Shape of the 2Fo-Fc and anomalous density maps suggests a single hydrated K ⁺ . There are 4 cations here, with distances between K ⁺ s ~3.0 Å. I do not think it is a possible arrangement.
	66		T	Also too many cations located at short distances from each other. Impossible arrangement.
	70		T	No anomalous density here.
	71		T	Too close to 153MgX.
	72		T	Too close to 156MgX.
	74		T	No anomalous density here. Too close to two Mg ²⁺ cations.
	81		T	81KT and 82KT are a single K ⁺ cation.
	86		T	86KT and 87KT are a single K ⁺ cation according to a single anomalous peak.
	90		T	There are too many closely positioned cations. Unlikely scenario.
	92		T	Here adjacent 571MgX is probably a K ⁺ while 92KT is a Mg ²⁺ according to the anomalous peak position.

	93		T	93K, 94K an 508MgX is probably a single hydrated K+.
	95		T	Too close to 265MgX. mg cation should be removed.
	96		T	This is not K+.
	101		T	This is not K+.
	34		V	Two Mg ²⁺ cations in close proximity.
	41		V	Close to 9MgX.
	42		V	Close to 307MgX.
	54		V	Close to 212MgX.
	58		V	Close to 194 MgX.
	76		V	Probably a single K+ cation instead of two here.
	78		V	Probably a single K+ cation instead of two here.
	131		V	Close to 262MgX.
	133		V	There are 5 cations here while the shape of the density suggests only 3 cations.
	138		V	No anomalous signal here. 4 cations here are probably a single hydrated Mg ²⁺ cation.
	139		V	4 cations here are probably a single hydrated K+.
	140		V	Too close to 297MgX.
	148		V	A whole bunch of cations here. 148KT and 100KT are probably a single K+.

REVIEWERS' COMMENTS:

Reviewer #2 (Remarks to the Author):

The authors have responded well to all of my concerns. I find this excellent study suitable for publication.

Reviewer #3 (Remarks to the Author):

The authors have addressed all my inquiries in the revised manuscript. Although I have not checked new cation assignments, I do see that the number of different cations has been updated and I trust the authors did good job of double-checking all of them.

I have a couple of minor suggestions:

- 1) Fig 1B, right panel. The Fe₄S₄ cluster could be a little bigger for better visualization.
- 2) Fig. 5 was cited ahead of Fig. 4C. I actually has not found a reference to Fig. 4C in the main text.

We thank the reviewers for taking time to read our manuscript for the second time.

REVIEWERS' COMMENTS:

Reviewer #2 (Remarks to the Author):

The authors have responded well to all of my concerns. I find this excellent study suitable for publication.

Reviewer #3 (Remarks to the Author):

The authors have addressed all my inquiries in the revised manuscript. Although I have not checked new cation assignments, I do see that the number of different cations has been updated and I trust the authors did good job of double-checking all of them.

I have a couple of minor suggestions:

1) Fig 1B, right panel. The Fe₄S₄ cluster could be a little bigger for better visualization.

Size of the cluster in the figure was adjusted.

2) Fig. 5 was cited ahead of Fig. 4C. I actually has not found a reference to Fig. 4C in the main text.

The figure was rearranged and reference added.